# The E3 ligases Itch and WWP2 regulate autoimmune neuroinflammation by controlling $T_H2$ to $T_H17$ cell conversion via interleukin-4-STAT5 axis in mice

Mei Zhao [1] ✉, Chao Zhang[1], Xin Zhang[1,2], Qingdian Mu[1], Qian Li[1] & Yun-Cai Liu [1,2] ✉

Multiple sclerosis (MS) is a neurodegenerative autoimmune disease primarily mediated by T helper 17 ($T_H17$) cells. We previously showed that Itch/WWP2 double knockout (DKO) T cells produce high levels of type 2 cytokines, driving spontaneous autoinflammation. Here, we report that DKO $T_H2$-high carrying autoantigen-specific TCR (2D2) develop atypical spontaneous experimental autoimmune encephalomyelitis (EAE), with CD4$^+$ T cells simultaneously producing IL-4 and GM-CSF, directly causing neuroinflammation. Unexpectedly, IL-4 deletion in DKO $T_H2$-high 2D2 mice exacerbates $T_H17$-driven classical EAE, indicating a $T_H2$ to $T_H17$ conversion. Furthermore, we show that the JAK3/STAT5 signaling pathway is critical for maintaining $T_H2$ lineage stability by modulating Blimp1 and c-Maf thereby suppressing $T_H17$ differentiation. Importantly, we find that this phenomenon can also be observed in dupilumab-treated patients with atopic dermatitis who develop psoriasis. Thus, our findings uncover the molecular antagonism and plasticity in the $T_H2$ and $T_H17$ cell programs and identify potential therapeutic targets for modulating $T_H2$ and $T_H17$ cell responses in autoimmune diseases.

Upon activation, naïve CD4$^+$ T cells differentiate into various T helper ($T_H$) cell subsets, including $T_H1$, $T_H2$, $T_H17$, and $T_{FH}$ cells, each playing distinct roles in responses against pathogens and tumors as well as in the development of inflammatory diseases[1,2]. $T_H2$ cells secrete interleukin-4 (IL-4), IL-5, and IL-13, thereby promoting B cell proliferation and IgE production[3], whereas $T_H17$ cells produce IL-17 and are important for defending against fungi and bacteria infection[4]. It has long been recognized that $T_H17$ cells are involved in autoinflammation, while $T_H2$ cells are responsible for driving allergic diseases such as asthma and atopic dermatitis. The potential involvement of $T_H2$ cells in autoimmune pathogenesis requires further investigation.

Experimental autoimmune encephalomyelitis (EAE) is the most commonly used experimental model for the human inflammatory demyelinating disease, multiple sclerosis (MS)[5]. Extensive evidence supports a central role for autoreactive CD4$^+$ $T_H$ cells in EAE pathogenesis, in particular $T_H1$ and $T_H17$ cells, which infiltrate into the central nervous system (CNS)[6,7]. In the CNS, the pathogenic $T_H17$ cells secrete the inflammatory cytokines IL-17, IFN-γ, and GM-CSF, leading to the activation and maturation of CNS-resident microglia and astrocytes, the recruitment of CNS-invading monocytes, and ultimately neuroinflammation and demyelination[8–10]. However, the specific role of $T_H2$ cells in EAE remains unclear. Although it was reported that myelin basic protein-specific $T_H2$ cells induce EAE in immunodeficient mice[11], it is generally thought that $T_H2$ cells play a protective role in autoimmune diseases by suppressing $T_H1$ cell differentiation[12,13]. On the other hand, $T_H17$ cells are also involved in the development of allergy[14,15]. As

[1]Institute for Immunology and School of Basic Medical Science, Tsinghua University, Beijing, China. [2]Tsinghua-Peking Center for Life Sciences, Tsinghua University, Beijing, China. ✉e-mail: zhaom18@tsinghua.org.cn; yuncai_liu@mail.tsinghua.edu.cn

previously reported, the frequency of $T_H17$ cells was higher in the blood of patients with atopic dermatitis[16]. Moreover, treatment with an anti-IL-4R antibody in allergic mice significantly induced $T_H17$ differentiation in the lungs, indicating a transition from $T_H2$ cells to $T_H17$ cells[17]. However, the cellular and molecular mechanisms that regulate the $T_H2$ and $T_H17$ cell plasticity remain to be elucidated.

The signal transducer and activator of transcription (STAT) family proteins are cytoplasmic transcription factors, which become phosphorylated and activated by multiple upstream kinases upon the cytokine receptor engagement, and then transcriptionally drive the differentiation of various $CD4^+$ T cells[18,19]. IL-4 is crucial for $T_H2$ cell differentiation. IL-4 interacts with the type I IL-4 receptor, consisting of IL-4Rα and common γ chain, and then activates the intracellular kinases JAK1 and JAK3, which not only induce the phosphorylation of STAT6 but also enhance the phosphorylation of STAT5[20,21]. The phosphorylated STAT6 upregulates GATA-3, the master regulator of $T_H2$ cells, which then enhances IL-4 production, promoting the development of the $T_H2$ lineage[22]. STAT5 also plays a key role in $T_H2$ cell differentiation by regulating IL-4 production[23]. Moreover, it has been reported that STAT5 is also involved in $T_H17$ cell differentiation, although its role is controversial[24–26]. In addition to GATA-3 and STAT proteins, several other transcription factors, including IRF4, c-Maf, and JunB, play significant roles during $T_H2$ cell differentiation[27–29].

We previously found that deletion of Itch and WWP2 (DKO) leads to spontaneous differentiation of $T_H2$ cells and type 2 autoinflammation[30]. In this study, we investigate the roles of $T_H2$ cells and IL-4 in the pathogenesis of neuroinflammation. We find that DKO 2D2 mice exhibit spontaneous atypical EAE driven by IL-4$^+$ and GM-CSF$^+$ $T_H2$ cells. However, deletion of IL-4 in DKO 2D2 mice exacerbates neuroinflammation and causes classical $T_H17$-driven EAE. We further establish that IL-4 promotes STAT5 phosphorylation, leading to Blimp1 upregulation and subsequent suppression of $T_H17$ cell differentiation. These findings uncover a previously unrecognized immunoregulatory role for $T_H2$ cells in neuroinflammation and offer insights into targeting the $T_H2$–$T_H17$ axis for treating autoimmune conditions.

## Results

### $T_H2$ cells are required for the development of spontaneous encephalomyelitis

We have previously demonstrated that $CD4^+$ T cells lacking the E3 ligases Itch and WWP2 secrete excessive type 2 cytokines and cause spontaneous lung inflammation in B6 mice[30]. However, it remained unknown whether these cells could recognize self-antigen and drive autoimmune responses. To address this issue, we crossbred the DKO-B6 mice with myelin oligodendrocyte glycoprotein (MOG) autoantigen-specific TCR transgenic 2D2 mice, which are commonly used to study the pathogenesis of MS[11]. At 16 weeks of age, the $Wwp2^{-/-}Itch^{f/f}Cd4$-Cre 2D2 mice (called "DKO mice" here) exhibited markedly lower body weight compared to their wild-type counterparts (Fig. 1a). DKO mice also showed higher titers of serum IgE than wild-type mice, while IgG1 levels were comparable between groups (Fig. 1b). Additionally, DKO mice displayed splenomegaly and enlarged inguinal lymph nodes (Fig. 1c). Flow cytometry analysis of the splenocytes revealed a lower frequency of $CD4^+$ T cells (Fig. 1d and Supplementary Fig. 1a) and an increased conversion of naïve T cells into effector T cells in DKO mice compared to wild-type mice (Fig. 1e). Splenic $CD4^+$ T cells from DKO mice produced more IL-4 than those from wild-type mice, implying a potential contribution of $T_H2$ cells to autoinflammation in DKO mice (Fig. 1f). However, GM-CSF production showed no significant difference in splenic $CD4^+$ T cells (Fig. 1f and Supplementary Fig. 1b).

We subsequently observed that over 80% of the DKO mice exhibited spontaneous atypical EAE symptoms, including ataxia, loss of balance, splayed legs, and tail rigidity (Fig. 1g, h). Histological analysis revealed that the cerebellum of the DKO mice exhibited neuroinflammation, characterized by lymphocyte infiltration and

demyelination (Fig. 1i). The infiltration of $CD45^{hi}CD11b^+$ myeloid cells and the microglia ($CD45^{mid}CD11b^+$) were increased in the CNS of DKO mice with atypical EAE symptoms (Fig. 1j–l). Although previous studies have generally linked atypical EAE development with neutrophil infiltration in the CNS[31,32], we found that infiltrating neutrophil levels were similar between DKO and wild-type mice (Supplementary Fig. 1c). Additionally, the proportion of infiltrating eosinophils was higher in the CNS of DKO mice than wild-type controls (Fig. 1m), indicating that type 2 immunity might be involved in the development of atypical EAE.

### $T_H2$-like $CD4^+$ T cells play an intrinsic role in the development of atypical EAE in DKO mice

$T_H2$ cells directly induce type 2 immune responses through the secretion of cytokines such as IL-4, IL-5, and IL-13, and indirectly by facilitating antibody production by B cells. To elucidate whether $T_H2$ cells were involved in the development of spontaneous atypical EAE in DKO mice, we analyzed the $CD4^+$ T cells in the CNS of DKO and wild-type mice. We observed a significant elevation in both the proportion and total number of $CD4^+$ T cells in DKO mice relative to their wild-type counterparts (Fig. 2a). Moreover, IL-4 production was markedly elevated in $CD4^+$ T cells from DKO mice (Fig. 2b), indicating that $T_H2$ cells had directly penetrated the CNS. Surprisingly, the $T_H2$-like $CD4^+$ T cells in DKO mice exhibited enhanced GM-CSF production, with an IL-4 and GM-CSF co-producing $T_H2$ population detected in the CNS (Fig. 2b). However, IL-17A production in $CD4^+$ T cells was comparable between groups, though IFN-γ production was increased in DKO $CD4^+$ T cells (Fig. 2c), possibly as a secondary effect of neuroinflammation.

To investigate whether $T_H2$ cells in DKO mice play an intrinsic role in atypical EAE development, we employed the passive EAE model (Fig. 2d). We isolated naïve $CD4^+$ T cells from DKO and wild-type mice and cultured them in vitro with anti-CD3 and anti-CD28 antibodies for 48 h. Activated $CD4^+$ T cells from DKO mice exhibited significantly higher levels of IL-4 and GM-CSF production in vitro, compared to those from wild-type mice (Supplementary Fig. 2a). We then adoptively transferred the stimulated T cells into immunodeficient $Rag1^{-/-}$ mice. $Rag1^{-/-}$ mice bearing $CD4^+$ T cells from DKO mice developed atypical EAE and exhibited significantly lower body weight compared to the control group receiving wild-type $CD4^+$ T cells (Fig. 2e, f). Consistent with the original phenotypes in DKO mice, infiltrating DKO $T_H2$ cells in $Rag1^{-/-}$ mice significantly increased IL-4 and GM-CSF production as well as IFN-γ production in the CNS (Fig. 2g, h). To elucidate the pathological role of GM-CSF in atypical EAE, we administered an anti-GM-CSF neutralizing antibody in a passive EAE model. Although anti-GM-CSF treatment did not affect the onset of EAE, it effectively alleviated the disease symptoms compared to the control group by day 24, indicating that GM-CSF exacerbates atypical EAE severity (Supplementary Fig. 2b, c). Furthermore, GM-CSF influenced the infiltration and activation of microglia and myeloid cells in the CNS (Supplementary Fig. 2d, e), but had no detectable effect on eosinophil or neutrophil populations (Supplementary Fig. 2f, g). Collectively, these findings demonstrate an intrinsic role of $T_H2$-like $CD4^+$ T cells in the development of atypical EAE.

### Loss of IL-4 in TKO mice exacerbates autoinflammation and potentiates classical EAE development

To further confirm the role $T_H2$ cells in the development of neuroinflammation, we deleted IL-4 expression in DKO mice (called "TKO mice" here) by crossing DKO mice with $Il4$ KO mice. Contrary to our expectations, TKO mice showed significantly lower body weight compared to DKO, $Il4$ KO, and wild-type mice as early as 8-week-old and exhibited higher mortality rates, with deaths occurring as early as 11 weeks of age (Fig. 3a, b). Both TKO and DKO mice developed splenomegaly and lymphadenopathy compared to $Il4$ KO and wild-type mice (Fig. 3c). Additionally, the frequencies of splenic $CD4^+$ T cells in TKO and DKO mice were lower than in wild-type and $Il4$ KO mice, but were comparable between DKO and TKO mice (Fig. 3d and

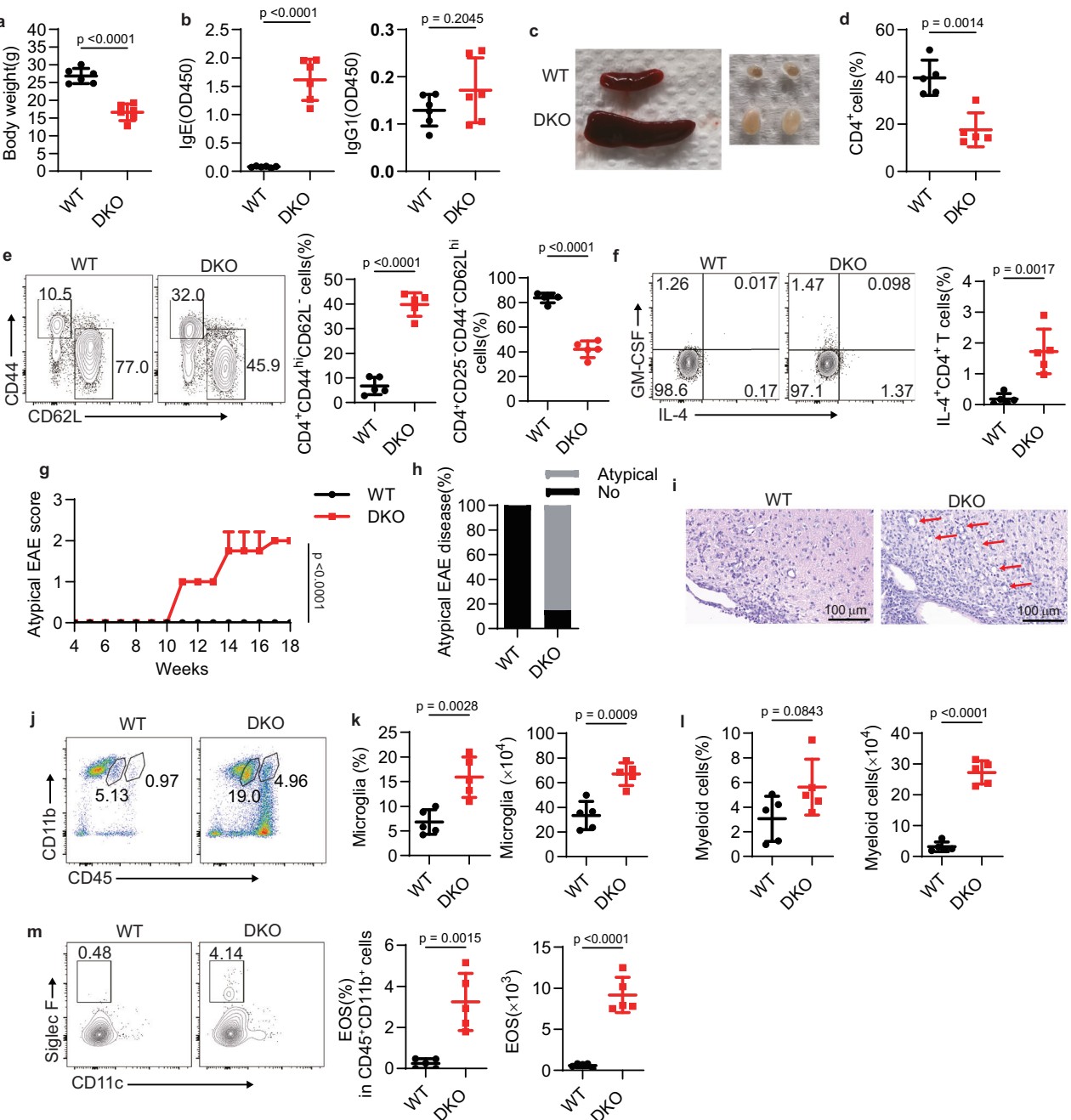

**Fig. 1 | DKO mice spontaneously develop atypical EAE disease. a** Body weight of 16-week-old female WT and DKO mice (key; $n = 6$ per group). Data are represented as mean ± SD. $p$ value was evaluated by unpaired two-tailed Student's $t$ test. **b** ELISA of IgE and IgG1 in the serum of 6- to 8-week-old mice as in (**a**) (key; $n = 6$ per group). Data are represented as mean ± SD. $p$ values were evaluated by unpaired two-tailed Student's $t$ test. **c** Representative images of spleens and lymph nodes in WT and DKO mice at 8-week-old. **d**, **e** Frequency of splenic CD4$^+$ T cells (**d**) and the effector-like (CD44$^{hi}$CD62L$^-$) CD4$^+$ T cells and naïve (CD25$^-$ CD44$^-$ CD62L$^{hi}$) CD4$^+$ T cells (**e**) from 6- to 7-week-old WT and DKO mice (key; $n = 5$ per group). Data are represented as mean ± SD. $p$ values were evaluated by unpaired two-tailed Student's $t$ test. **f** The production of IL-4 and GM-CSF in CD4$^+$ T cells from the spleen (key; $n = 5$ per group). Data are represented as mean ± SD. $p$ value was evaluated by unpaired two-tailed Student's $t$ test. **g** Clinical scores of atypical EAE based on ascending ataxia (key; $n = 8$ per group). Data are represented as mean ± SD. $p$ value on 18 weeks was evaluated by unpaired two-tailed Student's $t$ test. **h** Percentage of mice that developed atypical (ataxia and impaired balance) EAE symptoms. **i** H&E staining of brainstem lesions from 16-week-old WT and DKO mice. Scale bars, 100 µm. **j** Flow cytometry of CD45$^{hi}$ CD11b$^+$ myeloid cells and CD45$^{mid}$ CD11b$^+$ microglia in the CNS of WT and DKO mice. **k**, **l** Frequency and number of microglia (**k**) and myeloid cells (**l**) in the CNS of WT and DKO mice (key; $n = 5$ per group). Data are represented as mean ± SD. $p$ values were evaluated by unpaired two-tailed Student's $t$ test. **m** Frequency and number of eosinophils in the CNS of WT and DKO mice (key; $n = 5$ per group). Data are represented as mean ± SD. $p$ values were evaluated by unpaired two-tailed Student's $t$ test. Each symbol (**a**, **b**, **d**–**f**, **g**, **k**–**m**) represents an individual mouse. Data (**a**–**m**) are representative of at least three independent experiments.

Supplementary Fig. 3a). In splenic CD4$^+$ T cells, the proportion of effector CD4$^+$ T cells was higher in TKO mice compared with DKO mice, while naïve CD4$^+$ T cells were comparable between DKO and TKO mice (Fig. 3e).

We subsequently evaluated the clinical scores for EAE symptoms and found that deletion of IL-4 in DKO mice exacerbated the progression of EAE, as TKO mice exhibited tail paralysis as early as 6-week-old and underwent ascending paralysis, and the classical clinical scores

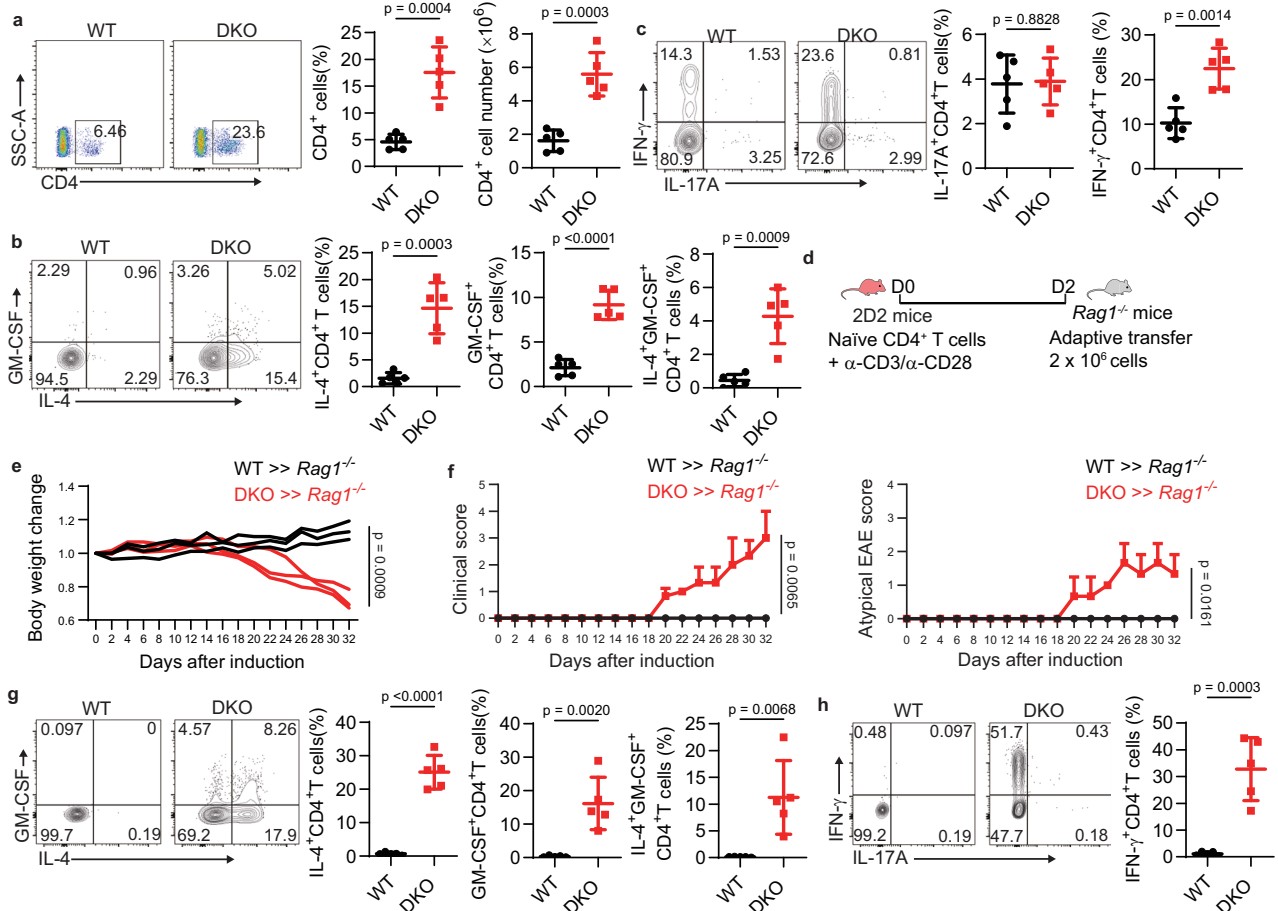

**Fig. 2 | T$_H$2 cells play an intrinsic role on atypical EAE development. a** Frequency and number of CD4$^+$ T cells in the CNS of WT and DKO mice (key; $n$ = 5 per group). Data are represented as mean ± SD. $p$ values were evaluated by unpaired two-tailed Student's $t$ test. **b** Frequency of IL-4$^+$ CD4$^+$ T cells, GM-CSF$^+$ CD4$^+$ T cells, and IL-4$^+$ GM-CSF$^+$ CD4$^+$ T cells in the CNS of WT and DKO mice (key; $n$ = 5 per group). Data are represented as mean ± SD. $p$ values were evaluated by unpaired two-tailed Student's $t$ test. **c** Frequency of IFN-γ$^+$ CD4$^+$ T cells and IL-17A$^+$ CD4$^+$ T cells in the CNS of mice as in (**b**) (key; $n$ = 5 per group). Data are represented as mean ± SD. $p$ values were evaluated by unpaired two-tailed Student's $t$ test. **d** The procedure for adoptive transfer in passive EAE. **e** Mouse body weight loss in the passive EAE model using WT and DKO CD4$^+$ T cells (key; each line represents an individual mouse, $n$ = 3

per group). $p$ value on Day32 was evaluated by unpaired two-tailed Student's $t$ test. **f** Classical and atypical EAE scores based on ascending paralysis and ataxia in the passive EAE model as in (**d**) (key; $n$ = 3 per group). Data are represented as mean ± SD. $p$ value on Day32 was evaluated by unpaired two-tailed Student's $t$ test. **g** The production of IL-4 and GM-CSF in the CNS of the passive EAE model as in (**d**) (key; $n$ = 5 per group). Data are represented as mean ± SD. $p$ values were evaluated by unpaired two-tailed Student's $t$ test. **h** The production of IL-17A and IFN-γ in the CNS of the passive EAE model as in (**d**) (key; $n$ = 5 per group). Data are represented as mean ± SD. $p$ value was evaluated by unpaired two-tailed Student's $t$ test. Each symbol (**a–c**, **e–h**) represents an individual mouse. Data (**a–c**, **e–h**) are representative of three independent experiments.

of TKO mice were consistently higher than DKO mice (Fig. 3f). Histological analyses showed lymphocyte infiltration, demyelination, and mononuclear cuffing in the spinal cord of TKO mice (Fig. 3g). Compared with DKO mice, demyelination in the spinal cord was more severe in TKO mice (Fig. 3h). In the CNS, the frequency of CD4$^+$ T cells was higher in TKO mice compared with wild-type and *Il4* KO mice, though no significant difference was observed between DKO and TKO groups (Fig. 3i). Additionally, the expression of the T$_H$17 cell-associated master transcription factor RORγt was significantly increased in TKO CD4$^+$ T cells, suggesting that IL-4-deficient DKO T cells had converted into RORγt$^+$ T$_H$17-like cells, particularly within the CNS (Fig. 3j, k). To validate these findings, we employed the passive EAE model (Fig. 2d). After induction, the body weight significantly decreased in mice of TKO group compared to DKO group by day 26 (Fig. 3l). The clinical scores representing classical EAE were also significantly higher in the TKO group than in DKO, *Il4* KO, and wild-type groups, while the atypical EAE scores were detected in the DKO group (Fig. 3m). These data collectively indicate that IL-4 negatively regulates classical EAE development.

## IL-4 controls T$_H$2 and T$_H$17 cell polarization with tissue specificity

To elucidate how IL-4 deficiency influences the pathogenesis of EAE, we examined cytokine production of CD4$^+$ T cells in the CNS across various spontaneous EAE mice. We observed a significant reduction in IL-4 and GM-CSF production in TKO mice compared to DKO mice, with a marked increase in IL-17A production in TKO mice (Fig. 4a, b). These findings indicate that deficiency of IL-4 promotes the conversion of Itch/WWP2-deficient CD4$^+$ T cells from IL-4$^+$ T$_H$2 cells into IL-17A$^+$ T$_H$17 cells. Additionally, no significant differences were observed in the frequencies of IFN-γ$^+$ CD4$^+$ T cells and IFN-γ$^+$ IL-17A$^+$ CD4$^+$ T cells among wild-type, *Il4* KO, DKO, and TKO mice (Fig. 4b), which effectively excluded the influence of T$_H$1 cells. Consequently, loss of IL-4 in CD4$^+$ T cells facilitates the shift from T$_H$2 to T$_H$17 cells, exacerbating the progression of neuroinflammation from atypical to classical EAE.

To investigate whether the effect of IL-4 on the T$_H$2 to T$_H$17 transition is tissue-specific, we also analyzed cytokine production in CD4$^+$ T cells from various tissues. Consistent with previous data, splenic CD4$^+$ T cells from DKO mice showed increased IL-4 production

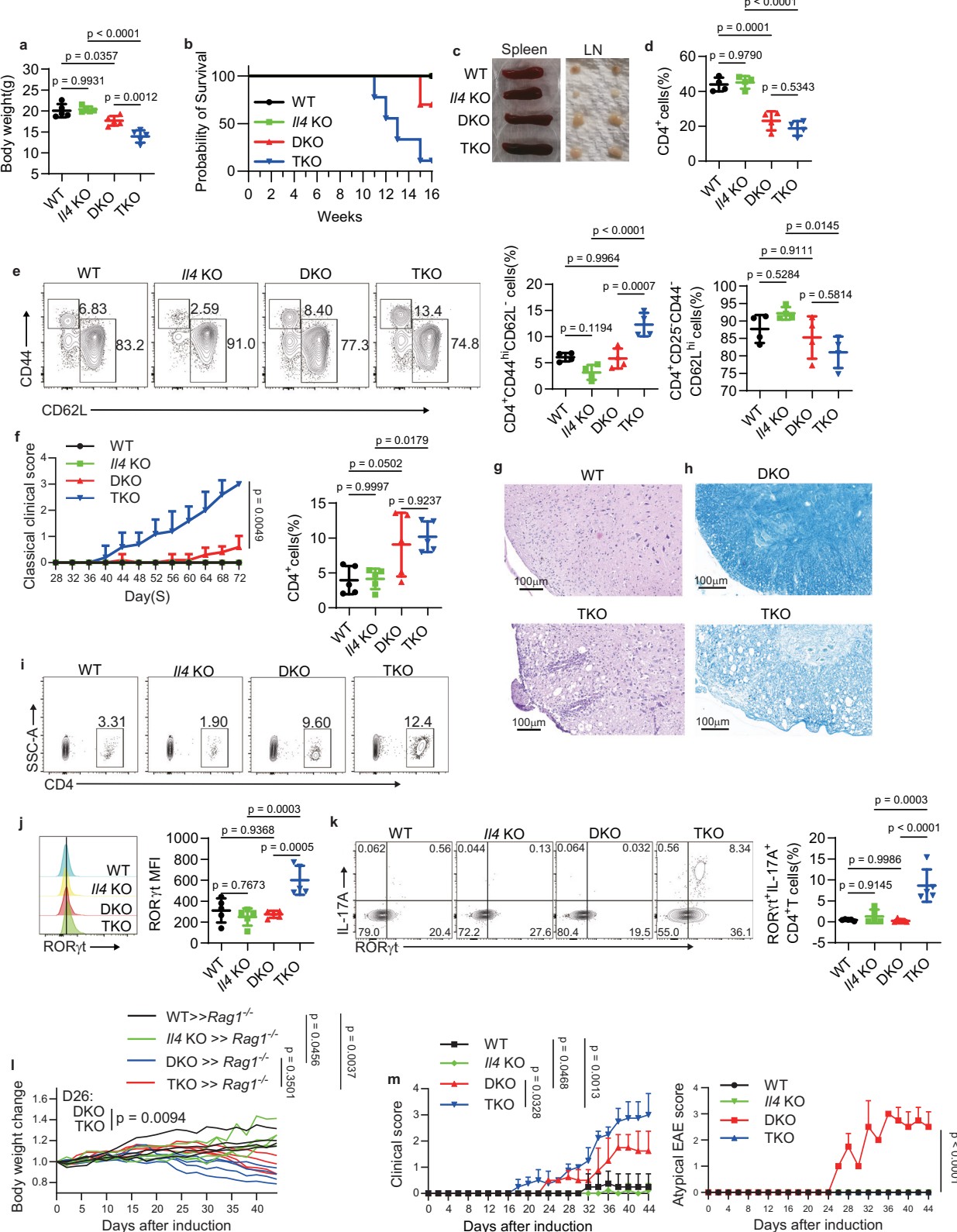

compared to other groups, while GM-CSF and IFN-γ production were similar in these four groups (Fig. 4c, d). Splenic CD4$^+$ T cell in TKO mice produced slightly more IL-17A than those in DKO mice (Fig. 4c). Given that T$_H$17 cells drive mucosal inflammation, we also analyzed the CD4$^+$ T cells in the colon. Over 10% of colon CD4$^+$ T cells produced IL-17A; however, there were no significant variations in IL-17A production among the four groups (Fig. 4e), which might be due to the immune

microenvironment. Additionally, IL-4 and GM-CSF production in CD4$^+$ T cells were elevated in DKO mice compared with other mice (Fig. 4f). Since T$_H$2 cells mediate allergic inflammation, we analyzed cytokine production in lung and skin. The frequency of CD4$^+$ T cells was similar in the lungs of 6- to 8-week-old wild-type, *Il4* KO, DKO, and TKO mice (Supplementary Fig. 3b). DKO mice showed enhanced production of IL-4 and GM-CSF in CD4$^+$ T cells, while IL-17A production remained

**Fig. 3 | Loss of IL-4 in TKO mice exacerbates the autoinflammation and classical EAE development. a** Body weight of 8-week-old female WT, *Il4*KO, DKO, and TKO mice (key; *n* = 5 per group). Error bars represent the mean ± SD. *p* values were evaluated by one-way ANOVA with Tukey's test adjusted for multiple comparisons. **b** Survival rate of mice as in (**a**) within 16 weeks (key; *n* = 10 per group). **c** Representative images of spleens and lymph nodes from 8-week-old mice as in (**a**). **d**, **e** Frequency of splenic CD4$^+$ T cells (**d**) and the effector-like (CD44$^{hi}$CD62L$^-$) CD4$^+$ T cells (**e**) and naïve (CD25$^-$ CD44$^-$ CD62L$^{hi}$) CD4$^+$ T cells (**e**) from 6- to 7-week-old mice as in (**a**) (key; *n* = 5 per group). Error bars represent the mean ± SD. *p* values were evaluated by one-way ANOVA with Tukey's test adjusted for multiple comparisons. **f** Classical clinical scores based on ascending paralysis in mice as in (**a**) (key; *n* = 5 per group). Error bars represent the mean ± SD. *p* value on Day72 was evaluated by unpaired two-tailed Student's *t* test. **g** H&E staining of spinal cord lesions in WT and TKO mice. Scale bars, 100 μm. **h** LFB staining of spinal cord lesions in DKO and TKO mice. Scale bars, 100 μm. **i** Frequency of CD4$^+$ T cells in the CNS of mice as in (**a**) (key; *n* = 5 per group). Error bars represent the mean ± SD. *p* values were evaluated by one-way ANOVA with Tukey's test adjusted for multiple comparisons. **j** The expression of RORγt in CD4$^+$ T cells in the CNS of mice as in (**a**) (key; *n* = 5 per group). Error bars represent the mean ± SD. *p* values were evaluated by one-way ANOVA with Tukey's test adjusted for multiple comparisons. **k** Frequency of RORγt$^+$ IL-17A$^+$ CD4$^+$ T cells in the CNS of mice as in (**a**) (key; *n* = 5 per group). Error bars represent the mean ± SD. *p* values were evaluated by one-way ANOVA with Tukey's test adjusted for multiple comparisons. **l** Mouse body weight loss in passive EAE model using WT, Il4 KO, DKO, and TKO CD4$^+$ T cells (key; each line represents an individual mouse, *n* = 4 per group). *p* values on Day44 were evaluated by one-way ANOVA with Tukey's test adjusted for multiple comparisons. **m** Classical clinical scores and atypical EAE scores in the passive EAE model as in (**l**) (key; *n* = 3 per group). Error bars represent the mean ± SD. *p* values of classical EAE score on Day44 were evaluated by one-way ANOVA with Tukey's test adjusted for multiple comparisons. *p* value of atypical EAE score on Day44 was evaluated by unpaired two-tailed Student's *t* test. Each symbol (**a**, **b**, **d**, **e**, **f**, **i**, **j**–**m**) represents an individual mouse. Data (**a**–**m**) are representative of three independent experiments.

unchanged (Supplementary Fig. 3c, d). In mice older than 12 weeks, the production of IL-17A was increased in lung TKO CD4$^+$ T cells compared to DKO mice (Supplementary Fig. 3e, f), a phenomenon we attribute to an indirect effect of the inflammatory milieu in aged mice. In post-auricular skin, production of cytokines associated with type 2 inflammation, such as IL-4, IL-13, and IL-25, was increased in DKO mice compared to the wild-type group (Supplementary Fig. 3g–i). Additionally, IL-17A production was slightly increased in TKO mice compared to DKO mice and was comparable to the wild-type group (Supplementary Fig. 3j). Collectively, these results indicate that the function of IL-4 in controlling T$_H$2 or T$_H$17 cell polarization is specific to the CNS.

### IL-4 Deficiency Drives T$_H$2-to-T$_H$17 Conversion In Vitro
To further confirm these in-vivo findings, we established an in-vitro T$_H$0 to T$_H$17-like cell culture system. Naïve CD4$^+$ T cells from wild-type, DKO, and TKO mice were cultured with anti-CD3 and anti-CD28 antibodies for 48 h, followed by supplementation with anti-IFN-γ, IL-6, and TGF-β. Cells were harvested on day 4 (Fig. 5a). An upregulated expression of GATA3 was observed in DKO cells compared with the wild-type and TKO cells under the T$_H$0, T$_H$2, and T$_H$0 to T$_H$17 polarizing conditions (Fig. 5b). However, this phenomenon was absent under T$_H$17 conditions (Fig. 5b). Furthermore, deficiency of IL-4 rescued RORγt expression in TKO cells under both T$_H$0 and T$_H$0 to T$_H$17 polarizing conditions (Fig. 5c). Further analysis of cytokine production showed that IL-4 production was significantly higher in DKO CD4$^+$ T cell group than in wild-type and TKO under T$_H$0, T$_H$2, and T$_H$0 to T$_H$17 conditions (Fig. 5d). Conversely, TKO CD4$^+$ T cells produced substantially more IL-17A under both T$_H$17 and T$_H$0 to T$_H$17 conditions (Fig. 5d). Given the relevance of pathogenic T$_H$17 (pT$_H$17) cells to neuroinflammation, we also investigated the differentiation of CD4$^+$ T cells both in pT$_H$17 and T$_H$0 to pT$_H$17 conditions (Supplementary Fig. 3k). In TKO mice, the naïve CD4$^+$ T cells also exhibited enhanced IL-17A production (Supplementary Fig. 3l). Similar findings were also detected in T$_H$2 to T$_H$17 conditions (Supplementary Fig. 3l). Collectively, these findings indicate that IL-4 regulates the bifurcation of naïve CD4$^+$ T cells into T$_H$2 or T$_H$17 under the T$_H$0 to T$_H$17 conditions.

### Loss of IL-4 reprograms T$_H$2 toward T$_H$17 transcriptional programs
To better understand how IL-4 controls T$_H$17 polarization, we conducted RNA sequencing (RNA-seq) to analyze the transcriptomes of naïve CD4$^+$ T cells and T$_H$17-like CD4$^+$ T cells cultured in vitro under T$_H$0 to T$_H$17 conditions across four groups (Fig. 5a). In naïve CD4$^+$ T cells, we observed 110 upregulated and 106 downregulated genes in TKO cells compared to DKO cells (Supplementary Fig. 4a). In polarized T$_H$17 cells from the TKO group, there were 1694 upregulated and 1393

downregulated genes relative to DKO T$_H$17-like cells (Fig. 6a), demonstrating significant transcriptional changes. Further, gene expression profiles of TKO and DKO groups at both the naïve and T$_H$17-like stages showed that differentially expressed genes were primarily enriched in pathways related to cytokine responses, T$_H$17 cell differentiation, IL-17 signaling, and the JAK-STAT pathway (Supplementary Fig. 4b, c). To elucidate the biological processes influenced by IL-4 in DKO and TKO CD4$^+$ T cells under T$_H$0 to T$_H$17 culture conditions, we conducted gene ontology (GO) enrichment analysis. As expected, differentially expressed genes were enriched in pathways related to T cell activation and differentiation, inflammatory response, and the cytokine signal responses (Fig. 6b).

Additionally, genes encoding transcription factors critical for T$_H$2 cell differentiation, such as *Gata3* and *Maf*, were expressed at higher levels in DKO polarized CD4$^+$ T cells compared to wild-type, *Il4* KO, and TKO cells (Fig. 6c). Conversely, transcription factors associated with T$_H$17 cell differentiation, such as *Rorc, Myc*, and *Tgif1*, were elevated in the TKO cell group (Fig. 6c). We further analyzed the cytokine and cytokine receptor genes that showed different expression between TKO and DKO cell group (Fig. 6d). The expression of *Il4* and *Il4ra* was upregulated in DKO cells, while *Il17a, Il17f*, and *Il22* showed higher levels in TKO cells (Fig. 6d). Collectively, these findings indicate that deletion of IL-4 in T$_H$2 cells substantially reduces T$_H$2 signature gene expression and promotes T$_H$17 signature gene expression. The trend of T$_H$2 to T$_H$17 conversion was progressive following IL-4 deletion. Additionally, *Il6ra* gene expression was significantly higher in both DKO and TKO cells than in wild-type and *Il4* KO cells (Fig. 6d). Furthermore, genes linked to pathogenic T$_H$17 cells, such as *Il23r, Ifng*, and *Il1r1*, were also elevated in the TKO group (Fig. 6d). To validate these findings, we conducted Gene Set Enrichment Analysis (GSEA) using RNA-seq data from TKO and DKO polarized T$_H$17-like cells against T$_H$-specific gene lists (Fig. 6e). The genes upregulated in T$_H$2 cells relative to T$_H$17 cells (GSE: 11924) were notably enriched in the DKO cell group.

Notably, a similar shift from T$_H$2 to T$_H$17-induced inflammation in atopic dermatitis under IL-4Rα blockade has also been reported, as shown in the onset of psoriasis subsequent to dupilumab therapy[33–36]. Re-analysis of a previous published human patient data revealed that the expression of genes associated with pathogenic T$_H$17 cells, including *Il17a, Il22, Il1β, Il23r*, and *Rorc*, was elevated in skin samples from dupilumab-induced psoriasis, whereas the expression of *Il4* and *Gata3* was increased in atopic dermatitis patients (Supplementary Fig. 4d, e)[37].

### IL-4 restricts T$_H$17 differentiation through promoting STAT5 activation
GO analyses further indicated enrichment of the TCR signaling pathway and the MAPK signaling pathway in polarized T$_H$17-like cells

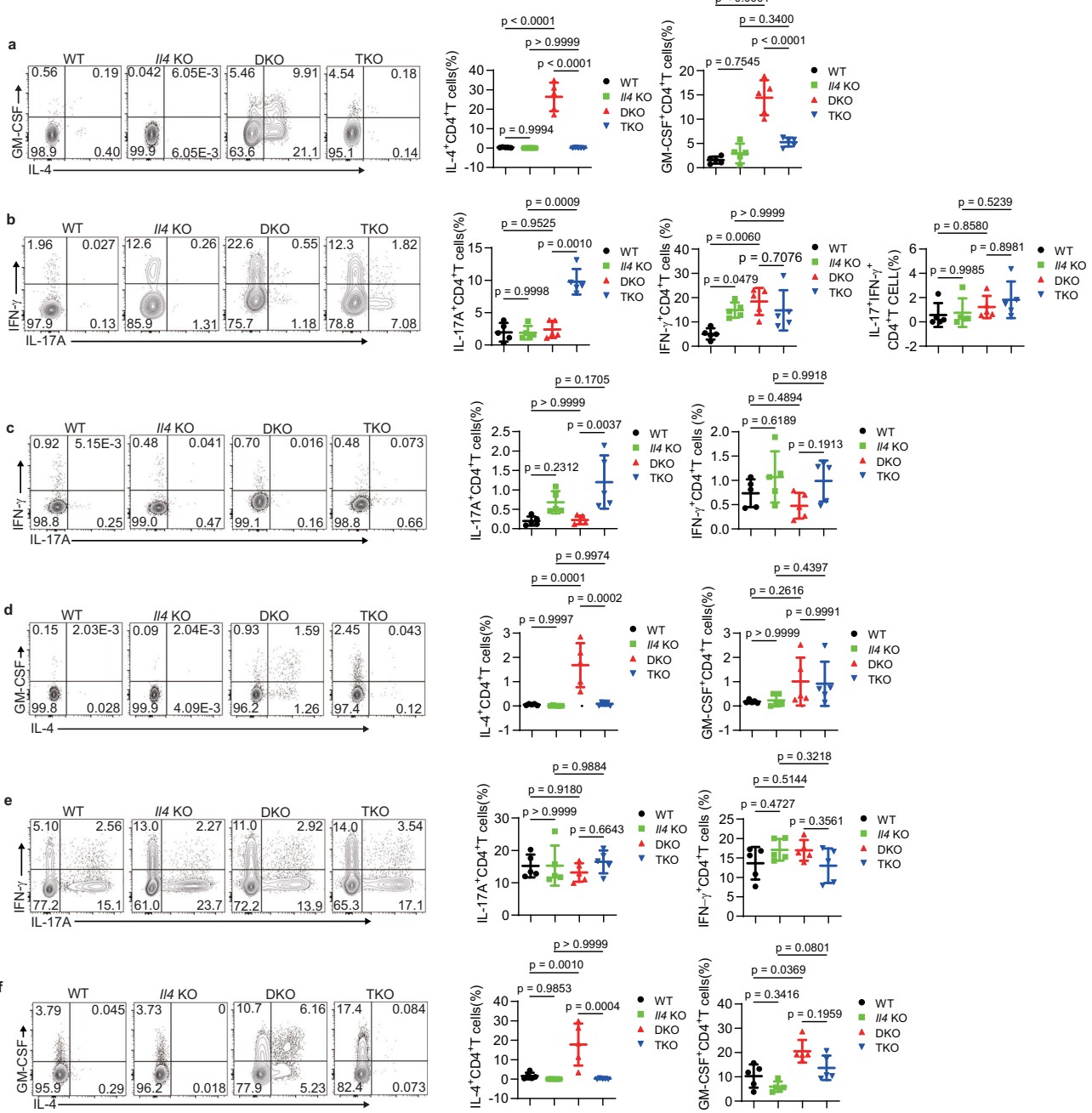

**Fig. 4 | IL-4 deficiency induces T$_H$2 conversion to T$_H$17 cells in CNS.**
**a, b** Frequency of cytokine production in CD4$^+$ T cells in the CNS of WT, *Il4* KO, DKO, and TKO mice (key; $n$ = 5 per group). Error bars represent the mean ± SD. $p$ values were evaluated by one-way ANOVA with Tukey's test adjusted for multiple comparisons. **c, d** Frequency of cytokine production in CD4$^+$ T cells in the spleen of mice as in (**a**) (key; $n$ = 5 per group). Error bars represent the mean ± SD. $p$ values were evaluated by one-way ANOVA with Tukey's test adjusted for multiple comparisons. **e, f** Frequency of cytokine production in CD4$^+$ T cells in the colon of mice as in (**a**) (key; $n$ = 5 per group). Error bars represent the mean ± SD. $p$ values were evaluated by one-way ANOVA with Tukey's test adjusted for multiple comparisons. Each symbol (**a**–**f**) represents an individual mouse. Data (**a**–**f**) are representative of three independent experiments.

(Fig. 6b). We assessed ERK1/2 activation using western blot analysis and found that the phosphorylation levels of ERK1/2 were similar across all four groups (Supplementary Fig. 4f). Given our previous findings that Itch and WWP2 cooperate to regulate TCR signaling strength and influence CD4$^+$ T cell differentiation, we explored whether IL-4 regulates T$_H$17 cell differentiation via TCR signaling. We evaluated the activation of TCR-proximal signaling components and observed that IL-4 deficiency did not affect the phosphorylation of ZAP70, LAT, and PLCγ1 in CD4$^+$ T cells upon anti-CD3 plus anti-CD28 stimulation (Supplementary Fig. 4g).

Multiple studies have shown that the JAK/STAT signaling pathway plays a critical role in T cell polarization[38]. STAT5 and STAT6 promote T$_H$2 cell differentiation and STAT3 is the major promotor of T$_H$17 cell differentiation[39–42]. Our gene enrichment analysis showed that the JAK/STAT signaling pathway was enriched in both naïve CD4$^+$ T cells and polarized T$_H$17-like cells from TKO mice compared with those from DKO mice (Fig. 6b and Supplementary Fig. 4b, c). We next explored whether IL-4 modulates STAT activation in CD4$^+$ T cells. We stimulated the splenocytes with MOG$_{35-55}$ for 48 h in vitro and detected the phosphorylation of STAT3, STAT5, and STAT6 in CD4$^+$ T cells using

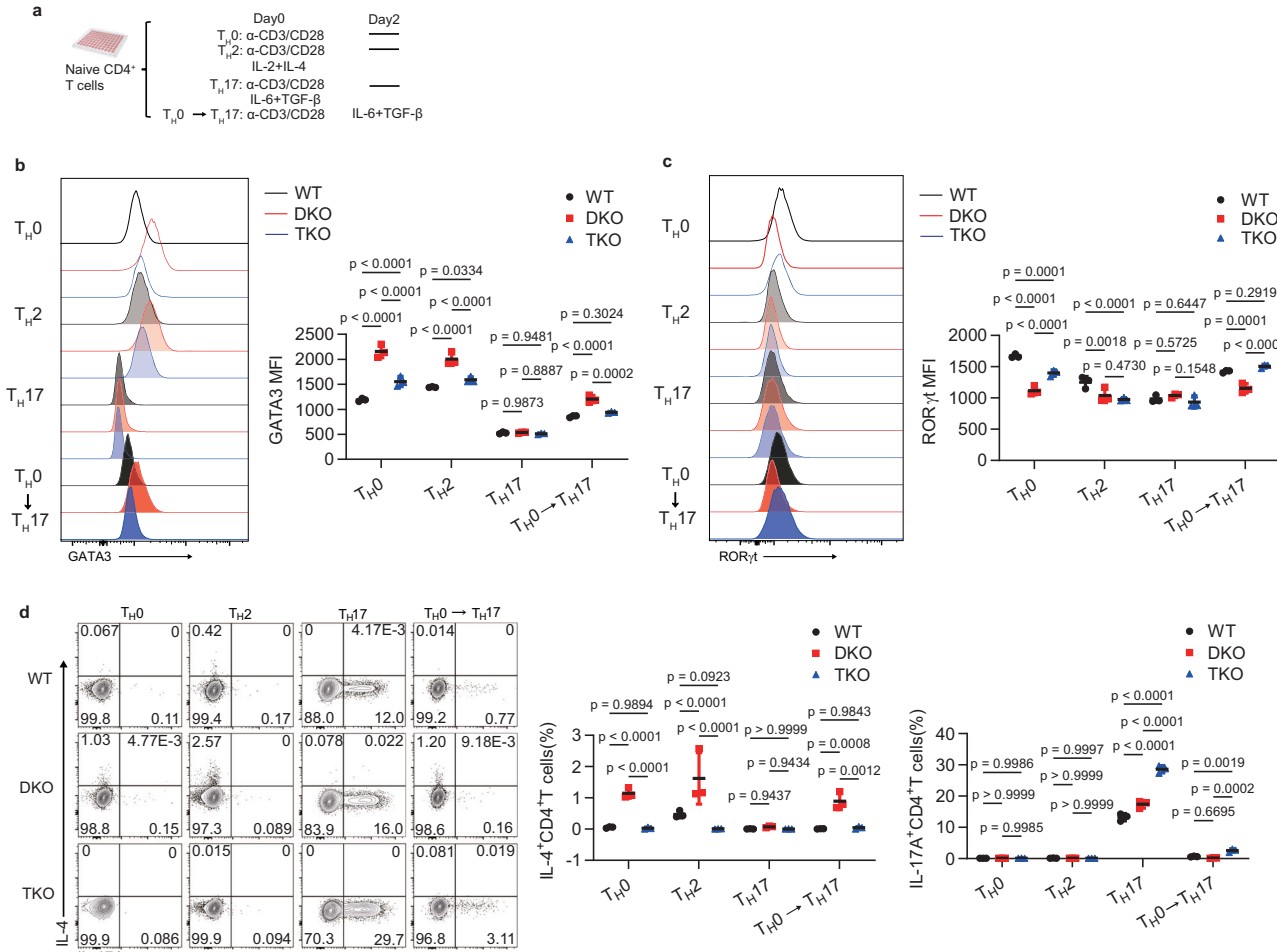

**Fig. 5 | IL-4 deficiency induces $T_H2$ conversion to $T_H17$ cells in vitro. a** The procedure for in-vitro T cell differentiation. **b, c** Expression of GATA3 (**b**) and RORγt (**c**) in WT, DKO, and TKO CD4+ T cells cultured in vitro as in (**a**) (key; $n = 3$ per group). Error bars represent the mean ± SD. $p$ values were evaluated by two-way ANOVA with Tukey's test adjusted for multiple comparisons. **d** Frequency of IL-4

and IL-17A production in the CD4+ T cells from WT, DKO, and TKO mice, which were cultured in vitro as in (**a**) (key; $n = 3$ per group). Error bars represent the mean ± SD. $p$ values were evaluated by two-way ANOVA with Tukey's test adjusted for multiple comparisons. Each symbol (**b–d**) represents one individual biological replicate. Data (**b–d**) are pooled from three independent biological replicates.

flow cytometry. Unstimulated DKO CD4+ T cells exhibited higher phosphorylation of STAT6 than in wild-type, *Il4* KO and TKO T cells (Fig. 6f). However, the phosphorylation level of STAT6 in DKO cells was similar to that of wild-type counterparts upon stimulation (Fig. 6f). IL-4 deletion significantly reduced STAT6 phosphorylation in *Il4* KO and TKO CD4+ T cells, consistent with the previous findings that IL-4 is critical for STAT6 activation (Fig. 6f)[43]. We observed that the phosphorylation levels of STAT5 was similar across all four groups in unstimulated CD4+ T cells (Fig. 6g). However, DKO CD4+ T cells exhibited higher phosphorylation of STAT5 than TKO cells after stimulation (Fig. 6g), indicating that IL-4 regulates STAT5 phosphorylation, either directly or indirectly. Since phosphorylation of STAT3 is the indicator of $T_H17$ differentiation, we found that phosphorylation of STAT3 was higher in TKO CD4+ T cells than in *Il4* KO and DKO cells after stimulation with the MOG$_{35-55}$ peptide (Supplementary Fig. 4h), consistent with the above-mentioned findings that the TKO CD4+ T cells preferentially differentiated into $T_H17$-like cells. Overall, these findings suggest that IL-4 regulates the activation of STAT3, STAT5, and STAT6.

To investigate whether phosphorylation of STAT proteins is the direct downstream regulator of IL-4 that restricts $T_H17$ differentiation, we treated naïve CD4+ T cells from DKO and TKO mice with STAT protein inhibitors under $T_H0$ to $T_H17$ conditions (Fig. 5a). As expected, treatment with specific STAT6 inhibitor AS1517499 blocked IL-4

production in DKO CD4+ T cells in a dose-dependent manner, whereas promoted IL-17A production (Supplementary Fig. 4i). However, AS1517499 treatment in DKO CD4+ T cells significantly increased IL-17A production relative to TKO cells (Supplementary Fig. 4i), suggesting that STAT6 is not the direct downstream regulator of IL-4 in controlling $T_H17$ cell differentiation.

Besides promoting $T_H2$ cell differentiation, STAT5 plays a crucial regulatory role in $T_H17$ cell differentiation[24]. We treated naïve CD4+ T cells with STAT5-IN, a potent and highly specific STAT5β inhibitor[44]. Increasing concentrations of STAT5-IN treatment led to a dramatic upregulation of IL-17A production in both DKO CD4+ T cells and TKO cells compared to the control group (Fig. 6h). Furthermore, At lower concentrations (1 μM), STAT5-IN moderately increased IL-17A production in both cell groups (Fig. 6h). In the 1 μM and 10 μM STAT5-IN treated groups, TKO CD4+ T cells produced more IL-17A than DKO counterpart (Fig. 6h). Additionally, high concentrations of STAT5-IN treatment counteracted the IL-4-mediated inhibition of IL-17A production in both DKO and TKO cells (Fig. 6h). STAT5-IN also caused a significant, dose-dependent reduction in IL-4 production in DKO CD4+ T cells (Fig. 6h). Taken together, these findings demonstrate that IL-4 reduces IL-17A production in DKO CD4+ T cells under $T_H0$ to $T_H17$ culture conditions by directly regulating STAT5 activation, rather than STAT6. While the treatment with STAT5-IN did not affect GATA3

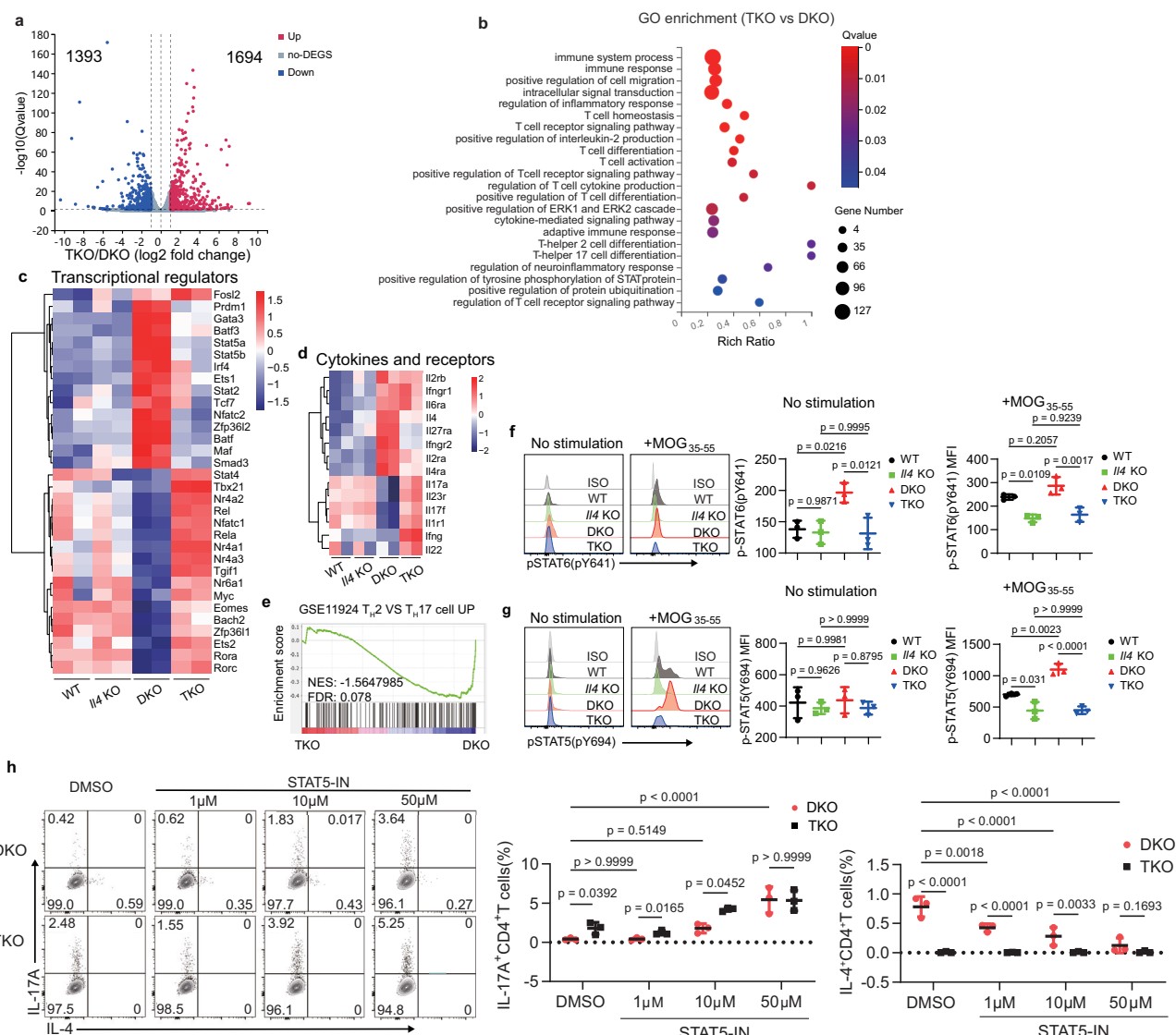

**Fig. 6 | IL-4 controls STAT5 phosphorylation and T$_H$17 cell differentiation.**
**a** Differential gene-expression profiles (x-axis, log2 (fold change)) and significance
(y-axis, −log10 (Q value)) between DKO and TKO CD4$^+$ T cells cultured in vitro and
stimulated for 4 h with PMA and ionomycin. **b** GO enrichment analysis of the dif-
ferentially expressed genes between TKO and DKO polarized T cells as in (**a**).
**c**, **d** Heatmap showing the transcription factors (**c**) and cytokines production (**d**)
associated with T$_H$2 and T$_H$17 cells in vitro cultured CD4$^+$ T cells from WT, *Il4* KO,
DKO, and TKO mice (key; n = 2 per group). **e** GSEA plots showing the enrichment of
differentially expressed genes expressed from the comparison between DKO and
TKO CD4$^+$ T cells as in (**a**). **f** Phosphorylation level of STAT6 in splenocytes from WT,
*Il4* KO, DKO, and TKO mice before and after MOG$_{35-55}$ stimulation (key; n = 3 per

group). Error bars represent the mean ± SD. p values were evaluated by one-way
ANOVA with Tukey's test adjusted for multiple comparisons. **g** Phosphorylation
level of STAT5 in splenocytes as in (**f**) (key; n = 3 per group). Error bars represent the
mean ± SD. p values were evaluated by one-way ANOVA with Tukey's test adjusted
for multiple comparisons. **h** Frequency of IL-4 and IL-17A production in CD4$^+$ T cells
from DKO and TKO mice cultured in vitro under T$_H$0 to T$_H$17 conditions with the
treatment of STAT5-IN (key; n = 3 per group). Error bars represent the mean ± SD. p
values were evaluated by two-way ANOVA with Tukey's test adjusted for multiple
comparisons. Each symbol (**c**, **d**, **f**–**h**) represents one individual biological replicate.
Data are pooled from two (**a**–**d**) and three (**f**–**h**) independent biological replicates.

expression, RORγt expression increased only at high inhibitor con-
centrations in both DKO and TKO CD4$^+$ T cells (Supplementary Fig. 4j),
indicating that the function of STAT5 is RORγt and GATA3
independent.

## IL-4 promotes JAK3/STAT5 activation and restricts T$_H$17 differentiation

Next, we focused on delineating how IL-4 regulates STAT5 activation.
Cytokine-induced receptor dimerization leads to trans-
phosphorylation of JAK kinases and subsequent STAT activation[45].
We first investigated whether IL-4 deficiency affects cytokine receptor

expression on the cell surface. Splenic CD4$^+$ T cells from *Il4* KO and
TKO mice displayed reduced IL-4Rα (CD124) expression compared to
wild-type counterparts, while DKO CD4$^+$ T cells exhibited higher
expression of IL-4Rα than wild-type cells (Fig. 7a), consistent with
previous findings that IL-4 regulates the expression of IL-4Rα[46]. It is
generally accepted that IL-2 is one of the major cytokines inducing
STAT5 phosphorylation in CD4$^+$ T cells[47]. We assessed the expression
of IL-2Rα subunit CD25 to examine if IL-4 regulates IL-2 signaling and
found that there was no difference among these four groups (Fig. 7b),
indicating IL-4 deficiency does not impact IL-2R expression. IL-4 initi-
ally binds to IL-4Rα and recruits the common γ chain to form the type I

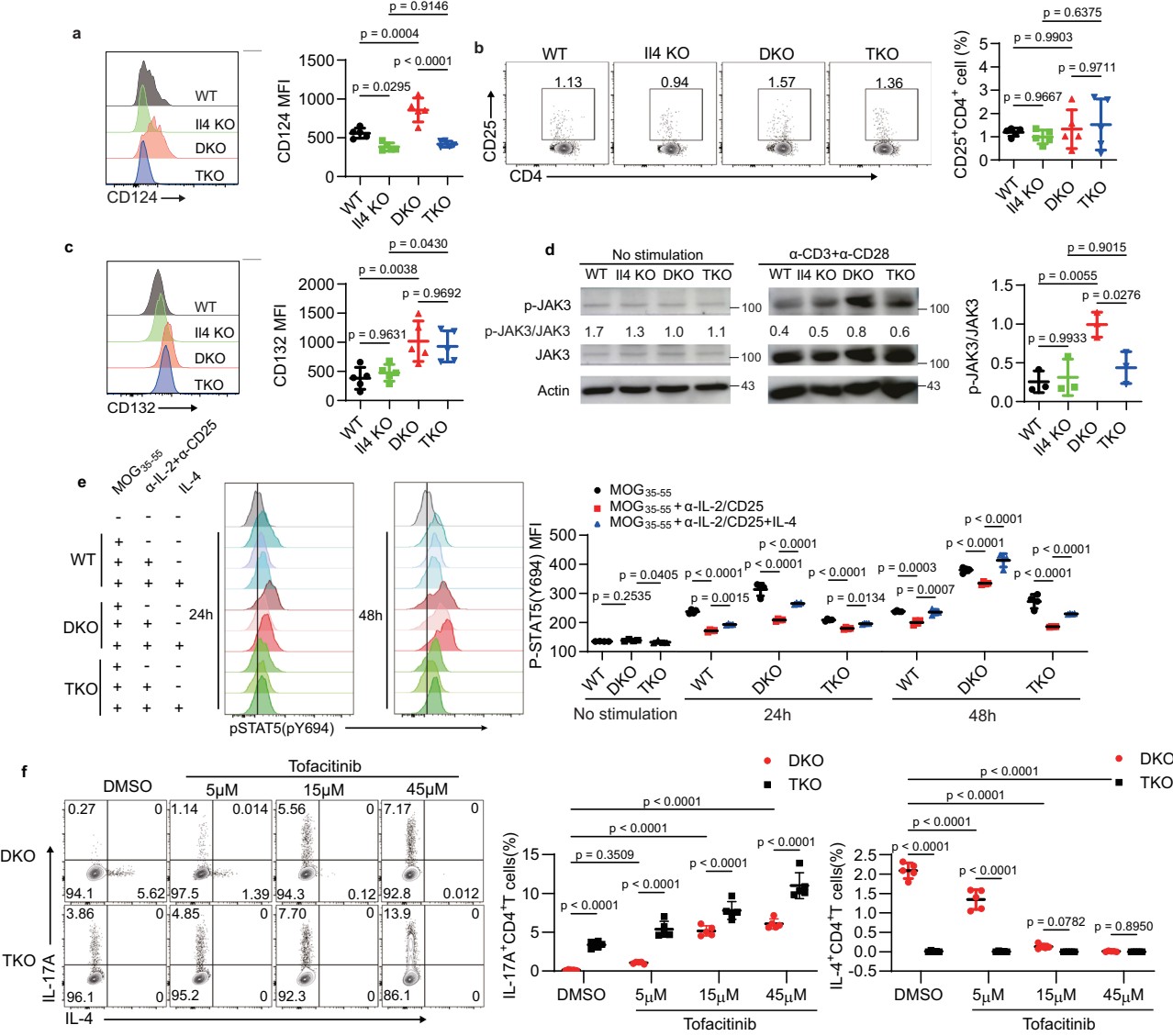

**Fig. 7 | IL-4R/JAK3/STAT5 signaling pathway controls T$_H$17 cell differentiation.**
**a** Expression of CD124 on splenic CD4$^+$ T cells from WT, *Il4* KO, DKO, and TKO mice (key; $n = 5$ per group). Error bars represent the mean ± SD. $p$ values were evaluated by one-way ANOVA with Tukey's test adjusted for multiple comparisons.
**b** Frequency of CD25 expression on splenic CD4$^+$T cells from mice as in (**a**) (key; $n = 5$ per group). Error bars represent the mean ± SD. $p$ values were evaluated by one-way ANOVA with Tukey's test adjusted for multiple comparisons. **c** Expression of CD132 on splenic CD4$^+$T cells from mice as in (**a**) (key; $n = 5$ per group). Error bars represent the mean ± SD. $p$ values were evaluated by one-way ANOVA with Tukey's test adjusted for multiple comparisons. **d** Immunoblot analysis of phosphorylated and total JAK3, in lysates of CD4$^+$ T cells sorted from mice as in (**a**) and stimulated for 20 h with anti-CD3 plus anti-CD28 (key; $n = 3$ per group); numbers below lanes indicate the ratio of phosphorylated protein to total protein. Error bars represent

the mean ± SD. $p$ values were evaluated by one-way ANOVA with Tukey's test adjusted for multiple comparisons. **e** Phosphorylation of STAT5 in splenic CD4$^+$ T cells from WT, DKO, and TKO mice stimulated with MOG$_{35-55}$, anti-IL-2 and anti-CD25, and IL-4 for 24 or 48 h (key; $n = 4$ per group). Error bars represent the mean ± SD. $p$ values were evaluated by two-way ANOVA with Tukey's test adjusted for multiple comparisons. **f** Frequency of IL-4 and IL-17A production in CD4$^+$ T cells from DKO and TKO mice cultured in vitro under T$_H$0 to T$_H$17 conditions with the treatment of tofacitinib (key; $n = 5$ per group). Error bars represent the mean ± SD. $p$ values were evaluated by two-way ANOVA with Tukey's test adjusted for multiple comparisons. Each symbol (**a–c**) represents an individual mouse. Each symbol (**d–f**) represents one individual biological replicate. Data (**a–c**) are representative of at least three independent experiments. Data are pooled from three (**d**), four (**e**), and five (**f**) independent biological replicates.

IL-4 receptor, initiating the activation of the downstream JAK3/ STAT5 signaling pathway[21]. We then found that common γ-chain (CD132) expression was slightly higher in DKO and TKO splenic CD4$^+$ T cells compared to their counterparts (Fig. 7c), which was associated with the activation of CD4$^+$ T cells. However, deletion of IL-4 did not affect CD132 expression in TKO cells compared to DKO cells (Fig. 7c).

Furthermore, to investigate whether deletion of IL-4 affects the activation of JAK3, we isolated the CD4$^+$ T cells from the spleens of wild-type, *Il4* KO, DKO, and TKO mice and stimulated in vitro with anti-CD3 and anti-CD28. JAK3 phosphorylation was similar in unstimulated

CD4$^+$ T cells across all four groups (Fig. 7d and Supplementary Fig. 5a). Following stimulation, DKO CD4$^+$ T cells exhibited higher JAK3 phosphorylation compared to wild-type, *Il4* KO, and TKO cells (Fig. 7d). Overall, these findings suggest that deletion of IL-4 does not affect common γ chain expression but regulates IL-4Rα expression and type I IL-4 receptor composition, thus influencing JAK3 and STAT5 activation in DKO CD4$^+$ T cells.

To confirm the role of IL-4 on STAT5 phosphorylation in DKO CD4$^+$ T cells, we activated splenocytes with MOG$_{35-55}$ peptide, blocked IL-2 signaling with anti-IL-2 and anti-CD25, and added IL-4 to induce IL-

4 signaling activation. Before $MOG_{35-55}$ peptide stimulation, STAT5 phosphorylation was comparable in wild-type, DKO, and TKO splenocytes, but increased over time after stimulation (Fig. 7e). Consistent with previous reports[48], anti-IL-2 and anti-CD25 treatment markedly reduced STAT5 phosphorylation in all three groups relative to $MOG_{35-55}$ stimulation alone (Fig. 7e), indicating that IL-2 signaling is crucial for STAT5 phosphorylation. Additionally, incubation of IL-4 to the anti-IL-2 and anti-CD25 treated groups markedly enhanced STAT5 phosphorylation (Fig. 7e), suggesting that IL-4 signaling is also important to induce the STAT5 phosphorylation. In particular, STAT5 phosphorylation in DKO CD4+ T cells treated with anti-IL-2 and anti-CD25 was significantly higher than in wild-type and TKO cells (Fig. 7e), indicating that IL-4 signaling predominantly induces STAT5 phosphorylation in DKO cells. In summary, these results highlight an essential role of IL-4 in driving STAT5 phosphorylation in DKO CD4+ T cells, through the γ chain of the type I IL-4 receptor and JAK3 signaling pathway.

Since the Janus kinase JAK3 is the upstream of STAT5, we investigated whether JAK3 inhibition promotes IL-17A production. We treated naïve CD4+ T cells from the spleens of DKO and TKO mice with tofacitinib, a JAK3 inhibitor[49], under $T_H0$ to $T_H17$ polarizing conditions (Fig. 5a). Tofacitinib treatment significantly enhanced IL-17A production in both DKO and TKO CD4+ T cells in a dose-dependent manner and simultaneously inhibited IL-4 production in DKO cells (Fig. 7f). These studies indicate that the deletion of IL-4 regulates the formation of type I IL-4 receptor and the activation of JAK3/STAT5 signaling pathway, thereby controlling $T_H17$ cell polarization.

Given the critical role of IL-6 signaling in $T_H17$ cell polarization, we also investigated whether the deletion of IL-4 affected the IL-6 signaling pathway, CD126 (IL-6Ra) expression on CD4+ T cells was similar among these four groups (Supplementary Fig. 5b). Moreover, JAK2 phosphorylation was undetectable in both unstimulated and stimulated cells among these groups (Supplementary Fig. 5c), indicating that IL-4 deletion does not directly affect IL-6 signaling. We also checked JAK1 phosphorylation. The unstimulated DKO and TKO CD4+ T cells showed slightly lower JAK1 phosphorylation (Supplementary Fig. 5d). After stimulation, DKO CD4+ T cells displayed higher JAK1 phosphorylation compared to wild-type, Il4 KO, and TKO cells, due to an increase in IL-4 receptor α signaling (Supplementary Fig. 5d).

### STAT5 prevents $T_H17$ cell polarization by inducing Blimp1 and c-Maf expression in DKO $T_H2$-like cells

To explore how STAT5 inhibits the differentiation of $T_H17$ cells in DKO $T_H2$-like cells, we performed genome-wide analysis of STAT5 occupancy in CD4+ T cells under $T_H0$ to $T_H17$ culture conditions using cleavage under targets and tagmentation (CUT&Tag). We integrated RNA-seq data comparing TKO and DKO CD4+ T cells with regions bound by STAT5 from the CUT&Tag data, identifying 167 genes induced by STAT5 and differentially expressed between DKO and TKO cells (Fig. 8a). Given previous reports that Blimp1 and c-Maf help control $T_H17$ cell polarization[50–52], we examined their genomic loci and observed significant STAT5 binding at the Prdm1 and Maf promoters in DKO cells (Supplementary Fig. 6a, b), but not at the Rorc or Il17a promoters (Supplementary Fig. 6c, d). Consistently, we analyzed two STAT5 binding motifs associated with each gene, and CUT&Tag-qPCR confirmed strong STAT5 occupancy at the Prdm1 and Maf promoters in DKO $T_H17$-like CD4+ T cells (Fig. 8b, c). Furthermore, the relative RNA levels of Prdm1 and Maf were also significantly increased in DKO CD4+ T cells compared with wild-type, Il4 KO, and TKO cells (Fig. 8d, e). Flow cytometric analysis also showed significantly higher expression of Blimp1 and c-Maf in DKO CD4+ T cells compared with the TKO counterparts (Fig. 8f, g), while single deletion of IL-4 dramatically reduced Blimp1 expression (Fig. 8f). These results collectively demonstrate that IL-4-STAT5 axis potently suppresses $T_H17$ cell differentiation, through direct binding of STAT5 to the Prdm1 and Maf promoters and driving Blimp1 and c-Maf expression in DKO CD4+ T cells.

To further confirm the role of STAT5 in suppressing $T_H17$ cell differentiation in DKO CD4+ T cells, we isolated the naïve CD4+ T cells from the four groups and infected them with retrovirus carrying either wild-type STAT5 or STAT5-CA, a constitutively activated form of STAT5, then induced differentiation of these infected cells under $T_H17$ conditions (Fig. 8h). As expected, STAT5-CA expression dramatically suppressed IL-17A production in these four groups (Fig. 8i), while IL-17A production slightly decreased in the wild-type STAT5 group (Fig. 8i), consistent with our finding that IL-4 modulates IL-17A production by regulating STAT5 phosphorylation. Furthermore, due to the strong $T_H17$-polarized conditions, GM-CSF production was limited (Fig. 8i) and IL-4 production was undetectable (Supplementary Fig. 7). To further elucidate the role of STAT5 phosphorylation in suppressing $T_H17$ cell differentiation in vivo, we utilized the 2D2 EAE adoptive transfer model (Fig. 8j). Mice receiving STAT5-CA-expressing T cells developed significantly milder disease than those receiving wild-type STAT5 or empty vector-transduced T cells (Fig. 8k, l). STAT5-CA-infected T cells infiltrating the CNS exhibited significantly lower frequencies of IL-17A production (Fig. 8m), indicating the suppressive role of STAT5 in IL-17A production in vivo. Conversely, the production of GM-CSF was significantly increased (Fig. 8m). There was no difference in the production of IFN-γ or in the frequency of IL-17A+ IFN-γ+ T cells in the CNS (Fig. 8m). Together, these results collectively validate STAT5 as a negative regulator of $T_H17$ cell polarization in DKO T cells in vivo.

## Discussion

In this study, we established a genetic model to dissect $T_H2$ cell functions in CNS autoimmunity using genetic manipulation. We observed that 2D2 mice lacking Itch and WWP2 exhibited atypical EAE symptoms, characterized by $T_H2$-dominant pathology. To confirm the role of $T_H2$ cells in the pathogenesis of atypical EAE, we knocked out the Il4 gene in DKO mice. Notably, IL-4 deletion exacerbated EAE symptoms and triggered a shift to classical EAE. Furthermore, deficiency of IL-4 induced a significant and spontaneous conversion of DKO $T_H2$ cells to TKO $T_H17$-like cells in both in-vitro and in-vivo conditions, indicating a dynamic balance between $T_H2$ and $T_H17$ cells. Mechanistic studies identified JAK3/STAT5 signaling as the molecular target: IL-4 sustained STAT5 phosphorylation, which promoted $T_H2$ stability while suppressing RORγt+ $T_H17$ differentiation.

$T_H2$ cells mediate type 2 immune responses, and their dysregulation drives the pathogenesis of allergic diseases. Comparatively, the pathogenesis of classical EAE involves $T_H1$ and $T_H17$ cells from the type 1 immune responses, which secrete pro-inflammatory cytokines, such as IFN-γ, IL-17A, and GM-CSF, and induce inflammation in the CNS. Our findings indicate that CD4+ T cells deficient in Itch and WWP2 produced more IL-4, leading to severe atypical EAE. Moreover, in the spontaneous atypical EAE model in DKO mice, we observed substantial infiltration of IL-4+ $T_H2$ cells and eosinophils in the CNS, as well as a significant increase in serum IgE titers in the DKO mice. These findings highlight the intrinsic contribution of $T_H2$ cells to the development of atypical EAE. Interestingly, we identified a unique population of IL-4 and GM-CSF double-positive $T_H2$ cells in the CNS of DKO mice, suggesting a novel immunoregulatory subset drive the pathogenesis of atypical EAE. We hypothesize that elevated GM-CSF production in DKO CD4+ T cells results from enhanced IL-4-induced STAT5 phosphorylation. GM-CSF is a key mediator of neuroinflammation known to recruit circulating myeloid cells and activate phagocytes in CNS[53,54] and the primary role of GM-CSF appears to be associated with disease severity, whereas the initial development of the disease is more likely linked to type 2 immunity. The potential roles of CNS-elevated GM-CSF in atypical EAE, particularly regarding the severity of atypical EAE, need further exploration.

IL-4 signaling deficiency impairs the IL-4Rα expression and the composition of type I IL-4 receptor complex, reducing the activation of

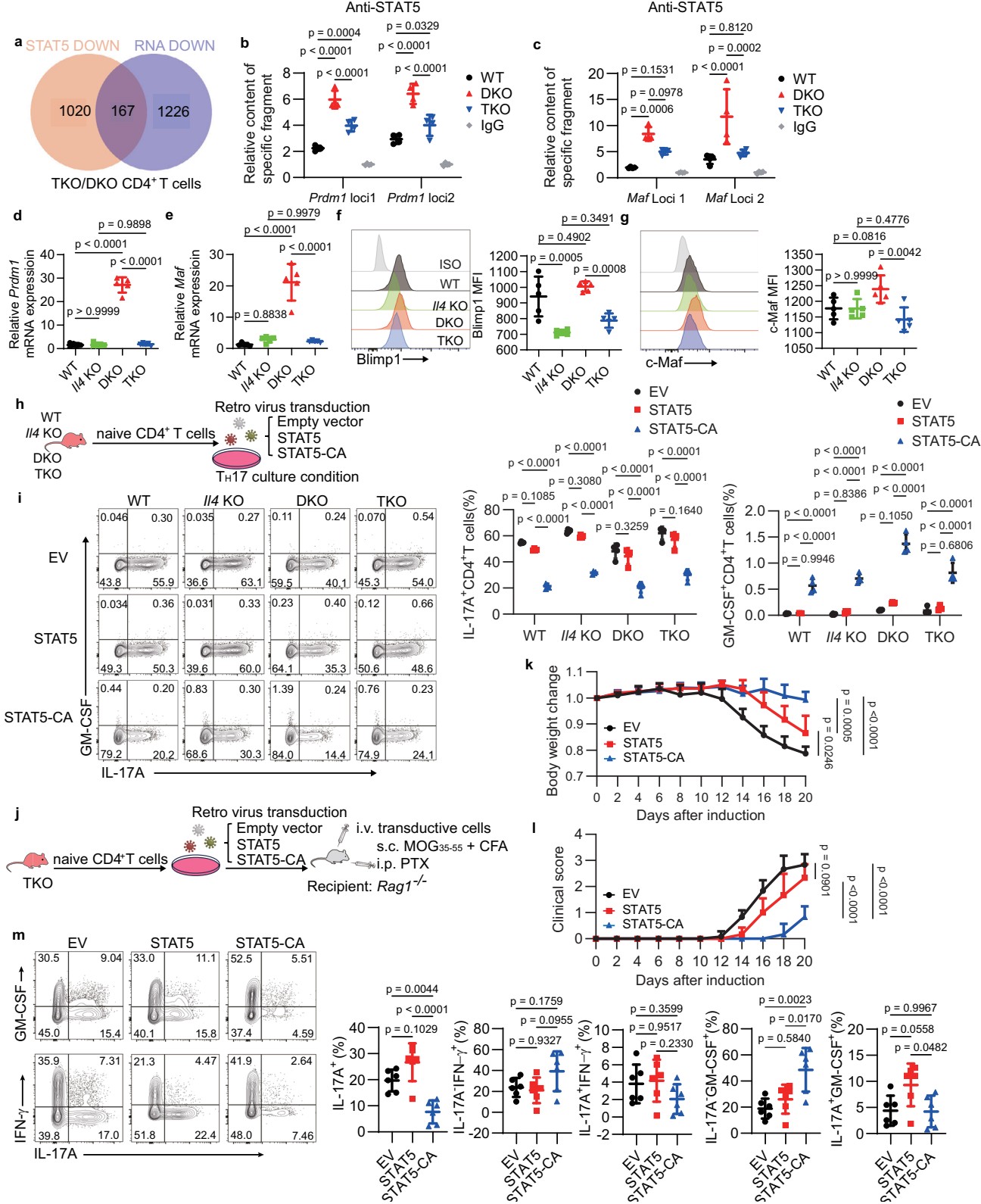

downstream mediators, such as STAT5 and STAT6, and leading to pathological $T_H17$-biased immune responses. Previous studies have shown that IL-2 signaling primarily activates STAT5[47], while our findings indicate that IL-4 signaling pathway is crucial for the function of STAT5 in $T_H2$ cells. IL-4 signaling alone sufficed to induce the phosphorylation of STAT5. Moreover, this study systematically examined the function of STAT5 in $T_H17$ cell differentiation by genetic and pharmacological approaches. Overexpression of activated STAT5

abolished IL-17A expression in vitro in $T_H17$-like cells and ameliorated EAE. In $T_{reg}$ cells, although STAT5 inhibits $T_H17$ cell polarization by directly binding to the *Rorc* and *Il17a* chromatin[24], we did not observe this in our study. Additionally, in DKO $T_H2$-like cells, STAT5 dimers translocate to the nucleus and directly bind to the *Prdm1* and *Maf* chromatin, thus inducing gene transcription. Previous studies have shown that Blimp1 inhibits $T_H17$ cell differentiation and Blimp1 and c-Maf synergistically regulate IL-10 production in human CD4+

**Fig. 8 | Stat5 directly binds to the Prdm1 and Maf and inhibits T$_H$17 cell differentiation in T$_H$2 cells. a** Venn diagram illustrating the overlap between genes that exhibit decreased STAT5 binding (orange) and genes that are transcriptionally downregulated (purple) in TKO CD4$^+$ T cells relative to DKO cells after in-vitro T$_H$0 to T$_H$17 polarizing culture. **b, c** CUT&Tag-qPCR analysis of the deposition of STAT5 at the *prdm1* (**b**) and *maf* (**c**) genes in CD4$^+$ T cells cultured in vitro under T$_H$0 to T$_H$17 conditions from WT, DKO, and TKO mice (key; *n* = 4 per group). Error bars represent the mean ± SD. *p* values were evaluated by two-way ANOVA with Tukey's test adjusted for multiple comparisons. **d, e** *Prdm1* (**d**) and *maf* (**e**) relative expression in CD4$^+$ T cells cultured in vitro under T$_H$0 to T$_H$17 conditions from WT, *Il4* KO, DKO, and TKO mice (key; *n* = 5 per group). Error bars represent the mean ± SD. *p* values were evaluated by one-way ANOVA with Tukey's test adjusted for multiple comparisons. **f, g** Expression of Blimp1 (**f**, *n* = 5 per group) and c-Maf (**g**, *n* = 5 per group) in CD4$^+$ T cells as in (**d** and **e**). Error bars represent the mean ± SD. *p* values were evaluated by one-way ANOVA with Tukey's test adjusted for multiple comparisons. **h** The procedure to generate STAT5 OE CD4$^+$ T cells from WT, *Il4* KO, DKO, and TKO mice and cultured in vitro under T$_H$17 conditions.

**i** Frequency of IL-17A and GM-CSF in CD4$^+$ T cells, which were infected with STAT5 or STAT5 mutation as in (**h**) (key; *n* = 4 per group). Error bars represent the mean ± SD. *p* values were evaluated by two-way ANOVA with Tukey's test adjusted for multiple comparisons. **j** The procedure to generate STAT5 OE CD4$^+$ T cells from TKO mice and induce adaptive transfer EAE model. **k** Mouse body weight loss in adaptive transfer EAE model as in (**j**) (key; *n* = 6 per group). Error bars represent the mean ± SD. *p* values on Day20 were evaluated by one-way ANOVA with Tukey's test adjusted for multiple comparisons. **l** Classical clinical scores in adaptive transfer EAE model as in (**j**) (key; *n* = 6 per group). Error bars represent the mean ± SD. *p* values on Day20 were evaluated by one-way ANOVA with Tukey's test adjusted for multiple comparisons. **m** Frequencies of cytokine production in CNS CD4$^+$ T cells from adaptive transfer EAE model as in (**j**) (key; *n* = 6 per group). Error bars represent the mean ± SD. *p* values were evaluated by one-way ANOVA with Tukey's test adjusted for multiple comparisons. Each symbol (**k**–**m**) represents an individual mouse. Each symbol (**b**–**g**, **i**) represents one individual biological replicate. Data (**k**–**m**) are representative of two independent experiments. Data are pooled from four (**b**, **c**, **i**) and five (**d**–**g**) independent biological replicates.

T cells[55,56]. We hypothesize that by enhancing the expression of Blimp1 and c-Maf, STAT5 controls T$_H$17 cell differentiation in DKO T$_H$2-like cells. Additionally, Blimp1 and c-Maf cooperate to limit T$_H$17 cell polarization. Collectively, we have shown that IL-4 is crucial in limiting T$_H$17 cell polarization in T$_H$2 cells, and that by enhancing the expression of Blimp1 and c-Maf, STAT5 suppresses the initiation of T$_H$17 cell differentiation. However, the precise mechanisms by which Blimp1 and c-Maf synergistically regulate T$_H$17 differentiation remain to be investigated.

While T$_H$2-biased responses are known to suppress T$_H$1 cell polarization, little is known about the balance between the differentiation of T$_H$2 cells and T$_H$17 cells. Previous reports suggested that classical T$_H$2 effector cells had the potential to produce IL-17 under ex vivo differentiation when stimulated with proinflammatory cytokines IL-1β, IL-6, and IL-21[57]. Conversely, RORγt deletion drives the conversion of mature T$_H$17 cells into T$_H$2-like cells[58]. Both reports, however, focus on plasticity in mature T cells at terminal stages. In contrast, our study reveals IL-4 acts as an early "switch" directing trans-differentiation between T$_H$2 and T$_H$17 lineages during naïve T cell activation under inflammatory conditions. The absence of IL-4 signaling in DKO CD4$^+$ T cells reduces the activation of STAT5, thereby promoting T$_H$17 differentiation specifically within the CNS of TKO mice. Similar findings have been observed in a mouse model of asthma in that treating asthmatic mice with an anti-IL-4 antibody increased IL-17A production in lung CD4$^+$ T cells[17]. Moreover, this plasticity extends beyond experimental models; analogous T$_H$2-to-T$_H$17 conversion was observed in atopic dermatitis patients treated with the IL-4Rα inhibitor dupilumab, which precipitated T$_H$17-driven psoriasis[33–36]. Our findings explain paradoxical autoimmune exacerbations during biologics therapy, and the mechanistic link between JAK3/STAT5 and T$_H$2/T$_H$17 balance offers potential therapeutic targets for multiple sclerosis, psoriasis, and related disorders. Although the murine EAE model is an invaluable tool for studying T helper cell dynamics in CNS autoimmunity, human MS exhibits chronic and heterogeneous pathology. Our findings elucidate fundamental mechanisms of T$_H$2 to T$_H$17 plasticity in autoimmune responses. However, the translational implications for human autoimmune diseases warrant further studies.

Notably, this cellular plasticity was restricted to the CNS and not observed in spleen or colon, indicating tissue- or antigen-specific regulation. We propose that the tissue-specificity is associated with the immune microenvironment. There are marked differences in the immune cells between the CNS and other tissues, such as T$_{reg}$ cells and innate immune cells, which produce different cytokines to regulate CD4$^+$ T cell responses. Besides, microbiota also plays a significant role in the differentiation of T$_H$17 cells in the intestine[59]. Therefore, it is possible that the microbiota may enhance TCR signaling in DKO CD4$^+$ T cells, overcoming the inhibitory function of IL-4 and thereby inducing the differentiation of T$_H$17 cells. It is known that oxygen levels vary among different tissues, and HIF, as a sensor of hypoxia, is involved in the regulation of T$_H$17 cell polarization[60]. Therefore, we speculate that the relatively oxygen-sensitive environment in the CNS may facilitate T$_H$17 cell differentiation compared to other tissues.

In summary, our data highlight the crucial and indispensable role of IL-4 in preserving the lineage stability and pathogenicity of mature T$_H$2 cells. These findings not only deepen our understanding of the role of IL-4 in T cell specification but also provide insights toward developing new therapeutic strategies for allergic/autoimmune diseases.

## Methods

### Sex as a biological variable
6–8 week-old male and female mice were examined in this study, with similar findings observed in both sexes. Female *Rag1$^{-/-}$* mice were utilized as recipient mice due to their increased susceptibility to EAE. Nevertheless, our findings are expected to be relevant to both sexes, since males develop EAE with a similar physiopathology.

### Mice and T cell lines
*Wwp2$^{-/-}$Itch$^{f/f}$Cd4*-Cre C57BL/6 mice have been previously characterized[30]. The 2D2 TCR transgenic mice were kindly provided by Dr. Chen Dong of Tsinghua University[11], Beijing, China. *Il4$^{-/-}$* mice (#002253) and *Rag1$^{-/-}$* mice (#002216) were acquired from the Jackson Laboratory. All mice were bred and housed under specific pathogen-free conditions in a temperature-controlled environment at 22–24 °C with a 12-h light/dark cycle. The animal experiments were performed following protocols approved by the Animal Care and Use Committee (number F16-00228-A5061-01 to Tsinghua University) and Institutional Review Board (20-LYC-1 to Liu Lab) of Tsinghua University. Experimental animals and control animals were bred separately. For euthanasia, the mice were euthanized by CO2 according to the approved protocols. The primary CD4$^+$ T cells involved in the in-vitro assays, retrovirus infection experiments, and passive EAE model were all derived from 2D2 transgenic mice.

### Antibodies
For T cell culture, antibodies to mouse CD3ε (clone 145-2C11, #BE0001-1, 3 μg/ml), CD28 (clone 37.51, #BE0015-1, 2 μg/ml), IFN-γ (clone R4-6A2, #BE0054, 10 μg/ml), IL-2 (clone JES6-1A12, # BE0043, 20 ng/ml) were from Bio X Cell; purified anti-mouse CD25 antibody were from Bio Legend (clone 3C7, # 101902, 1 μg/ml). For flow cytometry and ELISA, anti-mouse CD44-FITC (clone IM7, #45-0441-82, 1:500), anti-mouse Blimp-1-PE (5E7, #150005, 1:100), anti-mouse/human CD11b-BV421 (clone M1/70, #101236, 1:500), anti-mouse CD11c-PE/Cy7 (clone N418, #117318, 1:500), anti-mouse Ly6G/Ly6C(Gr1)-AF700 (clone 1A8,

#108422, 1:500), Purified anti-mouse IgE (clone RME-1, #406902, 1:2000), Purified anti-mouse IgG1 (clone RMG1-1, #406602, 1:4000) were from Bio Legend; anti-mouse CD4-eF450 (clone RM4-5,#48-0042-82, 1:500), anti-mouse CD25-PE (PC61.5, #25-0251-82, 1:500), anti-mouse CD62L-PE/Cy7 (clone MEL-14, # 25-0621-82, 1:500), anti-mouse CD126-PE (clone D7715A7,# 12-1261-80, 1:500), Anti-Mouse/Rat IL-17A-PE (clone 17B7, #12-7177-81, 1:400), anti-mouse IFN-γ-eF450 (clone XMG1.2, # 48-7311-82, 1:200), anti-mouse GM-CSF-PE/Cy7 (clone MP1-22E9, #25-7331-82, 1:100), anti-mouse RORγt-PerCP-Cy5.5 (clone B2D, #46698182, 1:100), anti-mouse/human c-Maf-PE (clone sym0F1, # 12-9855-41, 1:100) were from eBioscience; and anti-mouse Siglec-F-PE-CF594 (clone E50-2440, #562757, 1:1000), anti-mouse CD124-PE (clone mIL4R-M1, #552509, 1:500), anti-mouse IL-4-FITC (clone11B11, #557728, 1:100), anti-mouse GATA-3-FITC (clone L50-823, #560163, 1:100) were obtained from BD Biosciences. For Phosflow analysis, anti-mouse/human pSTAT3 at Tyr705-PE (eBioscience, clone LUVNKLA, #12-9033-42, 1:100), anti-mouse pSTAT5 at TyrY649-APC (BD Biosciences, clone 47, #612599, 1:100), anti-mouse pSTAT6 at TyrY641-PE (BD Biosciences, clone J71-773.58.11, #558252, 1:100). For immunoblot analysis, antibody to mouse Lck (clone 3A5, #sc-433, 1:2000) was from Santa Cruz Biotechnology; antibody to mouse ZAP70/Syk phosphorylated at Tyr319 and Tyr352 (clone 65E4, #2717, 1:1000), ZAP70 (clone D1C10E, #3165, 1:2000), PLCγ1 phosphorylated at Tyr783 (#2821, 1:1000), PLCγ1 (clone D9H10, #5690, 1:2000), Jak1 phosphorylated at Tyr1034 and Tyr1035 (clone D7N4Z, #74129, 1:1000), Jak1 (clone D1T6W, #50996, 1:2000), Jak2 phosphorylated at Tyr1008 (clone D4A8, #8082, 1:1000), Jak2 (clone D2E12, #3230, 1:2000), Jak3 phosphorylated at Tyr980 and Tyr981 (clone D44E3, #5031, 1:1000), Jak3 (clone D7B12, #8863, 1:2000), p44/42 MAPK (Erk1/2) phosphorylated at Thr202 and Tyr204 (clone D13.14.4E, #4370, 1:1000), p44/42 MAPK (Erk1/2) (clone 137F5, #4695, 1:2000) were from Cell Signaling Technology; antibodies to mouse LAT phosphorylated at Tyr191 (#07-278, 1:1000) and LAT (#06-807, 1:2000) were from Millipore; antibodies to Actin (clone AC-74, #A2228, 1:5000) was from Sigma-Aldrich; antibody to mouse Lck phosphorylated at Tyr394 (#PA5-37628, 1:1000) was from Thermo Fisher Scientific. For CUT&Tag analysis, antibody to mouse STAT5(D2O6Y, #94205, 1:50) was from Cell Signaling Technology; antibody to Rabbit IgG H&L(#ab6702, 1:100) was from Abcam. For anti-GM-CSF treatment, antibody to mouse GM-CSF (clone MP1-22E9, #HY-P99134, 20 mg/kg) and isotype control Rat IgG2a kappa (#HY-P990679, 20 mg/kg) were from MCE.

## T cell differentiation in vitro

2D2 TCR transgenic CD4$^+$ T cells were isolated from the spleen and peripheral lymph nodes using anti-CD4 microbeads (Miltenyi Biotec, clone REA604, #130-117-043, 1:10) according to the manufacturer's instructions. Subsequently, naïve CD4$^+$ T cells (CD4$^+$ CD25$^-$ CD62L$^{hi}$ CD44$^{low}$) were sorted using the BD FACSAria™ Fusion cell sorter (BD Biosciences). The cells were cultured in RPMI 1640 medium (Gibco, #C11875500BT) supplemented with 10% FBS (Gibco, #10091-148, 1:10), penicillin-streptomycin (Hyclone, #30010, 1:100), β-mercaptoethanol (Gibco, #21985023, 1:1000), and NEAA (Gibco, #11140050, 1:100) (T cell medium), and activated with plate-bound anti-CD3 (3 µg/ml) and anti-CD28 (2 µg/ml). T$_H$2 cell differentiation was induced with recombinant human IL-2 (Pepro Tech, #200-02-50, 10 ng/ml), mouse IL-4 (Pepro Tech, #214-14, 20 ng/ml), and anti-mouse IFN-γ (10 µg/ml). T$_H$17 cells were generated using a combination of human TGF-β (R&D Systems, #240-B-002, 1 ng/ml), mouse interleukin 6 (mIL-6) (Pepro Tech, #216-16, 20 ng/ml), and anti-mouse IFN-γ (10 µg/ml). To promote the differentiation from T$_H$0 to T$_H$17 conditions, naive CD4$^+$ T cells were initially activated with anti-CD3 and anti-CD28 for 48 h, and then changed the medium with T$_H$17-polarizing conditions. Molecular inhibitors, including AS1517499 (#S8685) and Tofacitinib (#S2789) from Selleck, as well as STAT5-IN (HY-101853) from MCE, were added to the culture medium as early as the T cells were stimulated with anti-

CD3/CD28 and/or cytokines. Four days later, cells were harvested for subsequent analyses.

## Retrovirus preparation and T cell infection

STAT5 and its mutants were cloned into a pMIG-IRES-GFP retroviral vector[61]. The plasmid and pCL-ECO packaging vector were transfected into PLAT-E (Platinum-Eco)cells using TransIT (Mirus, #MIR 2154) according to the manufacturer's instructions[62]. Approximately 4−6 h post-transfection, the medium was replaced with T cell medium, and the culture supernatant containing retroviral particles was collected at 24 and 48 h, then filtered through 45-µm filters. 2D2 TCR transgenic naïve CD4$^+$ T cells were sorted as previously described and activated with anti-CD3/CD28 for 20 h. Subsequently, the supernatant containing retroviruses, supplemented with polybrene (YEASON, #40804ES86, 8 µg/ml) and T$_H$17-polarizing cytokines, was added to the pre-activated CD4$^+$ T cells. The cells underwent spin infection at 500 × $g$ for 90 min at 37 °C and were then incubated for an additional 2 h. On day 2, the cells were reinfected and subsequently cultured under T$_H$17-polarizing conditions for 4 more days. After this period, the infected cells were harvested for further analysis.

## Adaptive transfer EAE induction

Naïve CD4$^+$ T cells were isolated from the spleen and lymph nodes of WT, *Il4* KO, DKO, and TKO 2D2 TCR transgenic mice and were stimulated in vitro using anti-CD3/CD28 for two days. Subsequently, 2−3 × 10$^6$ activated 2D2 TCR transgenic CD4$^+$ T cells were intravenously administered into female *Rag1*$^{-/-}$ recipient mice. Both body weight and disease scores were monitored daily. Classical EAE scores, ranging from 0 to 5, were assigned as previously described[63]. The presence of atypical disease manifestations, specifically balance issues and ataxia, was documented. Definitions of ataxia-related behaviors have been previously established[64]. Ataxic symptoms were evaluated on a cumulative 4-point scale, with each of the following physical manifestations assigned one point: splayed legs and tail rigidity, dragging of the trunk, and instability leading to lateral falls. For experiments involving Stat5 OE cells, these cells underwent transduction as outlined previously, followed by isolation of GFP-positive cells, which were then cultured under T$_H$17-polarizing conditions for an additional 24 h. Finally, 1 × 10$^6$ of these cells were collected and intravenously injected into female *Rag1*$^{-/-}$ recipient mice. For anti-GM-CSF treatment, a neutralizing anti-mouse GM-CSF antibody (20 mg/kg diluted in 0.2 ml of PBS) was injected intraperitoneally into mice every 2 days starting on day 2 following EAE induction. Control mice received rat IgG2a by the same injection protocols.

## Isolation of lymphocytes from CNS in EAE mice

At the peak of EAE disease progression, mice were euthanized and perfused intracardially with cold phosphate-buffered saline (PBS). The brain and spinal cord were subsequently extracted and washed extensively with cold PBS. The CNS tissues were finely minced before being mechanically dissociated through a 70 µm cell strainer. The resulting cell suspensions were enriched using a 40% Percoll (GE, #17089109) gradient, and the purified single-cell suspensions from the CNS were prepared for downstream flow cytometry analysis.

## Flow cytometry

Single-cell suspensions were initially stained with viability markers and surface antigen-specific antibodies. For intracellular cytokine detection, cells, whether cultured in-vitro or isolated from in-vivo experiments, were restimulated using phorbol 12-myristate 13-acetate (PMA; Sigma-Aldrich, #P8319, 50 ng/ml) and ionomycin (Sigma-Aldrich, #407952, 500 ng/ml), in conjunction with Golgi Stop (BD Biosciences, #554724, 1:1500) for 3−4 h prior to surface antigen staining. Subsequent steps included fixation and permeabilization (BD Biosciences, #554722) as directed by the manufacturer's guidelines. The stained

cells were then analyzed on a Fortessa5 (BD Biosciences) flow cytometer using FlowJo10 software, excluding dead cells via LIVE/DEAD® Fixable Violet Dead Cell Stain (Invitrogen, # L34955, 1:1600). Prior to phosphorylation staining (Phosflow), ex vivo CD4+ T cells were stimulated with specific cytokines for 15 min: mouse interleukin 6 (mIL-6) (Pepro Tech, #216-16,100 ng/ml) for STAT3 phosphorylation, recombinant human IL-2 (Pepro Tech, #200-02-50, 100 ng/ml) for STAT5 phosphorylation, and mouse IL-4 (Pepro Tech, #214-14, 100 ng/ml) for STAT6 phosphorylation. Post-stimulation, CD4+ T cells were fixed with fixation buffer (BD, #554655) for 30 min and permeabilized with methanol overnight at −20 °C. Following this, cells were washed with staining buffer (PBS with 1% FBS) and incubated with the respective phospho-STAT (pSTAT) antibodies as indicated before. Examples of gating strategies are provided in Supplementary Fig. 8.

### Brain and spinal cord histology
Before separating the brain and spinal cord, mice were perfused intracardially with cold PBS and then with 4% phosphate-buffered formalin. The whole CNS was fixed in 4% phosphate-buffered formalin overnight and embedded in paraffin. Subsequent sectioning (10 μm) and staining for hematoxylin and eosin (H&E) or luxol fast blue (LFB) were performed by Servicebio.

### Immunoblot
Purified CD4+ T cells (2 × 10^6) from the spleen and lymph nodes of 2D2 TCR transgenic mice before and after stimulation were lysed in 100 μL of RIPA lysis buffer (YEASON, #20101ES60) on ice for 30 min, followed by centrifugation at 3217 × g and 4 °C for 30 min. The supernatant was then transferred to a new tube and heated in SDS-PAGE loading buffer (CWbio, #CW0052, 1:5) at 98 °C for 10 min. The samples were subsequently loaded onto a 10% SDS-PAGE gel and electrotransferred to 0.45 μm polyvinylidene difluoride membranes (Millipore, #IPVH00010). These membranes were subjected to immunoblotting using specific primary antibodies and horseradish peroxidase–conjugated secondary antibodies. Detection was performed using an enhanced chemiluminescence system (Thermo Fisher Scientific, #34087). The grayscale values of protein bands were quantified using ImageJ software.

### RNA isolation and quantitative real-time PCR
For the quantitative analysis of gene expression via real-time PCR, total RNA was extracted from in vitro cultured cells using TRIzol reagent (Magen, #R4801-02), and cDNA was synthesized using a Superscript III First Strand Synthesis Kit (Invitrogen, #18080051) following the manufacturer's instructions. The real-time PCR assays were conducted in technical replicates using Taq Pro Universal SYBR qPCR Master Mix (Vazyme, #Q712-03) on a Light Cycler 2.0 system (Roche). Gene expression levels were normalized to the β-actin gene. The primers designed to amplify the target genes were as follows: for *Prdm1*, forward (5′-TTCTCTTGGAAAAACGTGTGGG-3′) and reverse (5′-GGAGCCG-GAGCTAGACTTG-3′); for *Maf*, forward (5′-GGAGACCGACCGCATCATC-3′) and reverse (5′-TCATCCAGTAGTAGTCTTCCAGG-3′).

### CUT&Tag-qPCR and library generation for sequencing
The CUT&Tag assay was conducted with several modifications using the Hyperactive Universal CUT&Tag Assay Kit for Illumina (Vazyme Biotech, #TD904). Initially, 5 × 10^5 primary T cells were isolated and coupled to ConA beads for 10 min. Subsequently, these bead-bound cells were incubated overnight at 4 °C in 50 μl of antibody buffer containing the primary antibody. Afterward, the secondary antibody was added and incubated for an additional hour. The cells were washed and incubated with 0.04 μM biotinylated-pA/G-Tnp for 1 h, followed by suspension in tagmentation buffer TTBL for another hour. The reaction was halted by the addition of Proteinase K and 10% SDS at 55 °C for 15 min. A spike-in mix (1 pg/10^5 cells) was incorporated for subsequent data normalization. Post the stopping step, tagmented DNA was

captured and isolated using streptavidin magnetic beads. Library construction was performed directly on the beads using the NEBNext Ultra DNA Library Prep Kit for Illumina (NEB, #E7370S). The completed CUT&Tag libraries were sequenced on the Illumina NovaSeq 6000 platform in PE150 mode by Novogene, Beijing, China.

For CUT&Tag-qPCR, the aforementioned CUT&Tag protocol was followed, except that after DNA extraction, the DNA-bound SA beads were resuspended and terminated with 5 μl of Stop buffer and heated at 95 °C for 5 min. The qPCR reactions utilized the purified DNA as templates and were conducted in triplicate. Gene-specific CUT&Tag-qPCR primers targeted 150–250 bp sequences surrounding the putative transcription factor-binding sites. The assays were accomplished with the following antibodies: anti-STAT5 antibody diluted 1:50; Anti-rabbit-IgG diluted 1:100.

### ATAC-seq library generation and sequencing
ATAC-seq was performed on approximately 1 × 10^5 viable cells using the Hyperactive ATAC-Seq Library Prep Kit for Illumina (Vazyme Biotech, #TD711) following the manufacturer's protocol with minor modifications. Briefly, cells were pelleted by centrifugation at 500 × g for 5 min at 4 °C and lysed in lysis buffer on ice for 5 min to isolate nuclei. Following nuclei collection by centrifugation under the same conditions, the pellet was resuspended in the transposome mix and incubated at 37 °C for 30 min to facilitate tagmentation. The reaction was terminated by adding stop buffer and incubating at 37 °C for 5 min. The tagmented DNA were subsequently purified using magnetic beads. The bead-bound DNA was washed twice with 80% ethanol and eluted in 20 μL of nuclease-free water. Library construction was performed using the NEBNext® Ultra™ DNA Library Prep Kit for Illumina® (NEB, #E7370S). Final libraries were sequenced on an Illumina NovaSeq 6000 platform (KaiTaibio, Hangzhou, China) utilizing paired-end 150-bp (PE150) reads.

### ATAC-seq and CUT&Tag data analysis
Raw paired-end sequenced reads were first cut for adaptor sequences and trimmed using trimmomatic (v.0.32). Then, cleaned fastq data were mapped to the mouse genome (mm10) using Bowtie 2 (v.2.4.2). Picard was used for marking and removing duplication. Samtools (v.1.11) was used for converting and sorting the SAM files into BAM format. Macs2 (v.2.1.0) was used for peak calling. Deeptools was used for normalizing signals from sorted BAM files via bamCoverage, and computeMatrix was used for calculating the overall signal distribution around the peak center called by macs2 using the "reference-point" model. Differential ATAC seq or Cut&Tag peaks were identified by DiffBind (v3.12.0). Data were annotated and visualized using Bioconductor-CHIP seeker (v.1.28.3) and IGV (v.2.11.9).

For CUT&Tag-qPCR, the relative proportion of target content was calculated using the ΔCt method. The Ct values of each CUT&Tag sample, whether antibody or IgG, can be normalized using its corresponding DNA spike-in as control: $\Delta Ct = Ct_{sample} - Ct_{DNA\ spike-in}$. Fold enrichment of the target region in each antibody sample was then determined relative to the IgG control: Fold enrichment = $2^{-\Delta Ct}$.

### RNA-seq and analysis
For the RNA sequencing analysis, total RNA was extracted from CD4+ T cells following stimulation with PMA and ionomycin using TRIzol reagent, based on the manufacturer's guidelines. The resulting RNA-seq libraries were constructed and subsequently sequenced by BGI Genomics on the DNBSEQ platform utilizing 50-bp paired-end reads. Differential expression analysis was performed using the negative binomial model implemented in Dr.Tom (v.2.0, BGI), which integrates DESeq2's statistical framework. Heat maps were generated to visualize these differences, with expression values computed as log2-transformed reads per kilobase of exon per million mapped reads (RPKM). Additionally, Gene Set Enrichment Analysis was conducted using the GseaPreranked method by GSEA(v.4.4.0), employing a

classic scoring scheme to assess the biological significance of observed expression patterns.

### Published microarray datasets analysis

The human patients skin microarray datasets are available from the Gene Expression Omnibus(GEO): GSE185764. Microarray probe ID was obtained from GEO. The CEL files from AD, DI-PSO patients and HC skin microarray data were imported into Transcriptome Analysis Console (v.4.0, Thermo Fisher Scientific), and then the gene expression list and DEGs generated by setting: Limma Anova, DABG (< 0.05) and the fold change of every gene (> 2 or <−2), together with their corresponding $p$ value or FDR $p$ value (Ben-jamini−Hochberg adjusted $p$ value) (< 0.05). Data were analysed in heatmap2 (v.3.2.0) and GraphPad Prism (v.10.0) for graphical representations.

### Statistical analysis

Statistical significance was assessed using Student's $t$-test, one-way or two-way analysis of variance (ANOVA), as appropriate, with GraphPad Prism (v.10.0). Each experiment was conducted a minimum of two to three times, yielding consistent results.

### Reporting summary

Further information on research design is available in the Nature Portfolio Reporting Summary linked to this article.

## Data availability

The raw sequence data reported in this paper have been deposited in the Genome Sequence Archive at National Genomics Data Center[65,66], China National Center for Bioinformation/Beijing Institute of Genomics, Chinese Academy of Sciences (https://ngdc.cncb.ac.cn/gsa/browse/CRA027859, accession code CRA027859). Previously published human patient skin microarray datasets used in this study is available in the Gene Expression Omnibus database (https://www.ncbi.nlm.nih.gov/geo/query/acc.cgi?acc=GSE185764, accession code GSE185764). All data are included in the Supplementary Information or available from the authors, as are unique reagents used in this Article. The raw numbers for charts and graphs are available in the Source Data file whenever possible. Source data are provided with this paper.

## Code availability

All data were analyzed using established, published pipelines with parameters detailed in the "Methods" section. No new custom code was generated for this study.

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

## Acknowledgements

We thank C. Dong of Tsinghua University for providing mouse strains for this study, and members of the Liu laboratory for support and discussion. We thank Jean-David Bouaziz from Saint Louis Hospital and Saint-Louis Research Institute for sharing the microarray gene expression data related to atopic dermatitis patients. We acknowledge the use of Sci-Draw (SciDraw. Scientific Drawings) for assistance in creating schematic diagrams included in this article. This work is supported by funding from the National Natural Science Foundation of China (NSFC82130051) to Y.-C.L., the Ministry of Science and Technology of China 2021 (YFC2300503) to Y.-C.L., the Tsinghua University-Xiamen Chang Gung Hospital Joint Research Center for Anaphylactic Disease to Q.L. and Y.-C.L. and the Tsinghua-Peking Center for Life Sciences to Y.-C.L.

## Author contributions

M.Z. designed and performed the experiments and analyzed data. C.Z. performed the bioinformatics analysis. X.Z. and Q.M. helped with the experiments and maintained mouse breeding colonies; M.Z., C.Z., Q.L., and Y.-C.L. interpreted the data and wrote the manuscript.

## Competing interests

The authors declare no competing interests.
