## [Peer Review File · Nature Communications]

The E3 ligases Itch and WWP2 regulate autoimmune neuroinflammation by controlling TH2 to TH17 cell conversion via interleukin-4-STAT5 axis in mice

Corresponding Author: Dr Mei Zhao

Version 0:

Reviewer comments:

Reviewer #1

(Remarks to the Author)

In this study by Zhao et al, the authors assessed the role of E3 ligases Itch and WWP2 in Th2 stability during autoimmune neuroinflammation. The authors found that mice containing MOG-specific TCR (2D2) and also lacking the E3 ligases Itch and WWP2 (DKO), developed spontaneous EAE. This EAE susceptibility 2D2-DKO mice was associated with increased IL-4 and GM-CSF producing T cells in the periphery and in the CNS. Interestingly, deletion of IL-4 in 2D2-DKO mice exacerbated EAE severity and this was associated with an elevated frequencies of Th17 cells and reduction in IL-4 producing Th2 cells. Mechanistic work performed by the authors revealed that the JAK3/STAT5 signaling pathway is critical for maintaining Th2 lineage stability by regulating the expression of Blimp1 and c-Maf, thereby preventing CD4+ T cells from differentiating into Th17 cells. The data appear of high quality, and support the conclusions. Overall, this is an interesting study that reveals the role of IL-4 in preserving the lineage stability of Th2 cells during CNS autoimmunity. Although the study is interesting there are some concerns that need to be addressed before the manuscript can be considered for publication.

To substantiate their claim that IL-4 limits the ability of DKO Th2 cells to become Th17 cells, the authors should perform Th17 polarization experiments with differentiated Th2 cells from DKO and TKO cells.

The authors should perform luxol fast blue staining in the CNS of DKO and TKO mice with EAE to validate their claim that TKO mice exhibit enhanced demyelination.

It's unclear why the profiling of STAT molecules was performed with T cells stimulated under MOG antigen while the gene expression profiling was performed with T cells polarized under Th0 and Th17 conditions.

Because RORγT can be expressed by non-Th17 cells, the authors should measure the frequencies of IL-17A+ RORγT double producers in the CNS of DKO and TKO mice with EAE to validate their claim that RORγT+ Th17-like cells are altered.

Th17 differentiation assays in Fig 4 were performed under non-pathogenic Th17 polarizing conditions. It's unclear if naïve T cells from TKO mice also exhibit enhanced ability to differentiate into pathogenic Th17 cells, which are relevant to EAE.

Although the work is generally rigorous, the full set of data represented in the figures is often not clear. Many of the panels say "n=#," but it's not clear if n refers to biological replicates from a single experiment vs independent experiment. Some figures do not mention the number of biological replicates used. This should be clarified in the figure legends.

Reviewer #2

(Remarks to the Author)

Building upon their previous results (Aki D et al., Nat Immunol 2018), Zhao et al. investigated the roles of Itch and WWP2 in CD4+ T cells within the context of neuroinflammation. Utilizing an extensive set of genetic mouse models and in vitro experiments, the authors found that MOG35-55-specific 2D2-TCR transgenic mice lacking Itch and WWP2 in CD4+ T cells exhibited atypical EAE symptoms, with immune responses characterized by a Th2-driven pathology, primarily mediated by

IL-4. Concerning downstream signaling of IL-4, the authors propose that activation of the IL-4 signaling pathway alone is sufficient for inducing the phosphorylation of STAT5. Furthermore, they systematically examined the function of STAT5 in Th17 cell differentiation, concluding that by enhancing the expression of Blimp1 and c-Maf, STAT5 suppresses the initiation of Th17 cell differentiation. As a clinical link to human disease, the authors re-analyzed skin sample datasets from dupilumab-associated psoriasis in individuals with atopic dermatitis and confirmed the upregulation of genes related to pathogenic Th17 cells.

This well-written manuscript presents a thorough analysis of the interaction between Th2 and Th17 differentiation, providing novel insights into the role of Th2-mediated immune responses in the context of autoimmunity. The authors have addressed a clinically relevant issue, and the methodology appears sound. However, a few points require clarification:

Major Criticism:

- In the introduction, the authors inaccurately describe the current understanding of the pathophysiology of multiple sclerosis (MS) and experimental autoimmune encephalomyelitis (EAE). Specifically, the authors assert that Th17 cells are the "key drivers of MS progression" (page 3, lines 41-22), which does not align with the current consensus on MS-related disease progression (Kuhlmann T et al., *Lancet Neurol* 2022). Additionally, they claim that "In the central nervous system (CNS), TH17 cells secrete the inflammatory cytokines IL-17, IFN- γ , and GM-CSF, which recruit polymorphonuclear cells from peripheral tissues to the CNS and lead to neuroinflammation" (page 3, lines 42-44). However, polymorphonuclear cells constitute only a minor population in MS lesions (Lassmann H, *Cold Spring Harb Perspect Med* 2018). Furthermore, the authors state that "TH2 cells are also implicated in the pathogenesis of MS" (page 3, lines 44-46). To date, convincing evidence for a significant role of Th2 cells has only been demonstrated in the context of EAE, not in human MS (Attfield KE et al., *Nat Rev Immunol* 2022). We recommend that the authors frame EAE as an appropriate model for studying the differential roles of T helper cells in shaping the autoimmune response, rather than focusing on MS pathophysiology, which is neither correctly described nor revisited in the discussion.

- Throughout the methods section, the authors do not consistently specify whether experiments were conducted using 2D2 T cells (e.g., pages 22-24). At times, it is unclear when 2D2 cells were employed. Please clarify when 2D2 cells are used to provide a cohesive statement that acknowledges their use throughout the manuscript, while also noting any exceptions.

- Additionally, it is crucial to ascertain whether the observed immune phenotype is exclusively mediated by 2D2 cells or also involves bystander cells. What is the percentage of 2D2 cells in the analysis, as shown in Figure 1D-F, Figure 2A-D, Figure 2G/H and Figure 3D/E. It is also unclear how many of the transferred cells are 2D2 cells (Figure 2D). Could these be measured using appropriate antibodies against the TCR alpha and beta chains or MOG-tetramers?

- In Figure 6D, the authors present an immunoblot analysis of phosphorylated and total JAK3. A quantification using densitometry would be good.

Minor Criticism:

- To investigate whether the effect of IL-4 on the Th2 to Th17 transition is tissue-specific, the authors analyzed cytokine production in CD4+ T cells from the colon and spleen (pages 8-10). However, given that other organs, such as the lung and skin, are commonly affected by Th2-mediated diseases (e.g., asthma and atopic dermatitis), it would be valuable to know whether these organs were also analyzed in the study.

- Considering previous data on the role of IL-4 in the context of neuroinflammation (Ponomarev ED et al., *J Neurosci* 2007), the authors tend to overstate the novelty of their findings in some instances. We advise them to moderate the wording, e.g., on page 19, lines 407-408 ("Unexpectedly, deletion of IL-4 exacerbated EAE symptoms and led to the development of classical EAE").

Reviewer #3

(Remarks to the Author)

Reviewer #4

(Remarks to the Author)

The manuscript entitled "The E3 ligases Itch and WWP2 regulate autoimmune neuroinflammation by controlling Th2 to Th17 cell conversion via interleukin-4-STAT5 axis" by Zhao et al investigates the roles of Itch, WWP2, and IL-4 in experimental autoimmune encephalomyelitis (EAE) using genetically deficient mouse models. The authors showed that deletion of Itch and WWP2 (DKO) leads to atypical EAE, characterized by neuroinflammation. Further deletion of the Il4 gene in the DKO mice (creating TKO mice) resulted in the conversion of Th2 cells to Th17 cells, driving the development of classical EAE. Mechanistically, they identify the IL-4R/JAK3/STAT5 signaling pathway as a key regulator that promotes Blimp1 expression, which subsequently inhibits Il17a expression. Although the study sheds some light on the phenomenon of atypical neuroinflammation and its molecular mechanism, I have the following concerns regarding the results presented.

1. The concept of “atypical EAE” remains unclear. While the authors attribute the neuroinflammation observed in DKO mice to IL-4, the co-expression of IL-4 and GM-CSF by DKO CD4+ T cells raises the possibility that GM-CSF, rather than IL-4, is the primary driver of neuroinflammation. GM-CSF is a well-established contributor to EAE pathogenesis, yet the authors have not explored its role in this phenomenon.

2. Although the mechanisms underlying Th2-to-Th17 plasticity identified in this study are novel, the advancement of knowledge is limited because the concept of CD4+ T cell plasticity is already well established in the field.

Version 2:

Reviewer comments:

Reviewer #1

(Remarks to the Author)

The authors addressed my concerns. The manuscript can be accepted for publication.

Reviewer #2

(Remarks to the Author)

All criticism has been appropriately and satisfactorily addressed. I have no further comments and congratulate the authors.

Reviewer #3

(Remarks to the Author)

Reviewer #4

(Remarks to the Author)

The authors properly addressed my concern.

Responses to Reviewers' Comments

We thank the reviewers for their careful reading, insightful comments and suggestions. These are valuable and helpful for us to improve the manuscript. We have addressed the questions point-by-point as described in the followings (in **blue** font). Furthermore, in the revised manuscript, edited texts are highlighted in **yellow** and all new or revised figures are highlighted with a **red box**.

Reviewer #1 (Remarks to the Author)

In this study by Zhao et al, the authors assessed the role of E3 ligases Itch and WWP2 in Th2 stability during autoimmune neuroinflammation. The authors found that mice containing MOG-specific TCR (2D2) and also lacking the E3 ligases Itch and WWP2 (DKO), developed spontaneous EAE. This EAE susceptibility 2D2-DKO mice was associated with increased IL-4 and GM-CSF producing T cells in the periphery and in the CNS. Interestingly, deletion of IL-4 in 2D2-DKO mice exacerbated EAE severity and this was associated with an elevated frequencies of Th17 cells and reduction in IL-4 producing Th2 cells. Mechanistic work performed by the authors revealed that the JAK3/STAT5 signaling pathway is critical for maintaining Th2 lineage stability by regulating the expression of Blimp1 and c-Maf, thereby preventing CD4+ T cells from differentiating into Th17 cells. The data appear of high quality, and support the conclusions. Overall, this is an interesting study that reveals the role of IL-4 in preserving the lineage stability of Th2 cells during CNS autoimmunity. Although the study is interesting there are some concerns that need to be addressed before the manuscript can be considered for publication.

We thank the reviewer #1 for the constructive feedback, which has significantly improved the quality of our revised manuscript. All raised concerns have been systematically addressed through expanded methodological validation, including T cell differentiation assays to further corroborate our conclusions.

To substantiate their claim that IL-4 limits the ability of DKO Th2 cells to become Th17 cells, the authors should perform Th17 polarization experiments with differentiated Th2

cells from DKO and TKO cells.

We thank the reviewer for this insightful suggestion. Accordingly, we performed the requested experiments.

As shown in the red box of Rebuttal Fig.1 (also supplementary Fig. 3j and k in the revised manuscript), we added new data about Th17 polarization experiments using *in vitro*-differentiated Th2 cells from DKO and TKO mice. The production of IL-17A was significantly increased in TKO CD4⁺ T cells compared to DKO cells.

Rebuttal Fig. 1, The production of cytokines in CD4⁺ T cells cultured *in vitro*. **a** The procedure for *in vitro* T cell differentiation. **b** Frequency of IL-4 and IL-17A production in the CD4⁺ T cells from WT, DKO, and TKO mice, which were cultured *in vitro* as in (a) (key; n= 3 per group). Each symbol represents one individual replicate.

The authors should perform luxol fast blue staining in the CNS of DKO and TKO mice with EAE to validate their claim that TKO mice exhibit enhanced demyelination.

We also thank the reviewer for raising this critical point. To address the comment, we have now performed Luxol Fast Blue (LFB) staining on spinal cord sections from both DKO and TKO mice with spontaneous EAE, specifically analyzing regions of demyelination (Rebuttal Fig. 2). This data has been incorporated into Fig. 3h and

described in lines 157-158 of the revised manuscript. We appreciate the opportunity to strengthen this aspect of our work.

Rebuttal Fig. 2, LFB staining of spinal cord lesions in DKO and TKO mice. Scale bars, 100µm.

It's unclear why the profiling of STAT molecules was performed with T cells stimulated under MOG antigen while the gene expression profiling was performed with T cells polarized under Th0 and Th17 conditions.

We appreciate the reviewer's observation regarding stimulation conditions. First, to mimic the in vivo CD4⁺ T cell activation upon MOG antigen recognition in the spleen, we stimulated splenocytes with MOG₃₅₋₅₅ for 48 hours to activate naïve T cells and measured STAT protein phosphorylation levels. This experimental design eliminated potential interference from exogenous cytokines on STAT phosphorylation levels.

Additionally, data from Fig. 5b demonstrate significant enrichment of STAT-associated phosphoregulatory proteins in the differentially expressed gene profiles between DKO and TKO CD4⁺ T cells polarized under Th0 to Th17 conditions, which are consistent with the conclusions drawn from both methods.

To directly address this reviewer's comment, we assessed STAT protein phosphorylation during Th0 to Th17 polarization. Phosphorylation levels of STAT5 and STAT6 were elevated in DKO CD4⁺ T cells compared to TKO cells, consistent with MOG stimulation data (Rebuttal Fig. 3). We believe this dual approach provides complementary insights into STAT protein phosphorylation in DKO and TKO cells. Due to the nature of this piece

of information is only additive to the already presented data, this new data will be used for reviewer's purpose only.

Rebuttal Fig. 3, the phosphorylation of STAT proteins in CD4⁺ T cells polarized under Th0 to Th17 conditions (key; n= 5 per group). Each symbol represents one individual replicate.

Because RORγT can be expressed by non-Th17 cells, the authors should measure the frequencies of IL-17A⁺ RORγT double producers in the CNS of DKO and TKO mice with EAE to validate their claim that RORγT⁺ Th17-like cells are altered.

We thank the reviewer for raising this important point. We agree that RORγT expression is not exclusive to Th17 cells. To rigorously validate alterations in Th17-like cells, we have now quantified IL-17A⁺ RORγT⁺ double-positive CD4⁺ T cells in the CNS of DKO and TKO mice with EAE. The new data are included in Rebuttal Fig. 4 (also Fig. 3k in the revised manuscript) and support our original conclusions.

Rebuttal Fig. 4, Frequency of RORγT⁺ IL-17A⁺ CD4⁺ T cells in the CNS of WT, I/4 KO, DKO, and TKO mice (key; n= 4 per group). Each symbol represents an individual mouse.

Th17 differentiation assays in Fig 4 were performed under non-pathogenic Th17 polarizing conditions. It's unclear if naïve T cells from TKO mice also exhibit enhanced ability to differentiate into pathogenic Th17 cells, which are relevant to EAE.

We appreciate the reviewer's insightful comment regarding the differentiation of pathogenic Th17 cells in our study. We agree that evaluating the potential of naïve T cells from TKO mice to differentiate into pathogenic Th17 cells is critical for understanding their role in EAE pathogenesis. To address this point, we conducted additional experiments using pathogenic Th17-polarizing conditions and included these data in the revised manuscript (the red box of Rebuttal Fig. 1, also supplementary Fig. 3j and k in the revised manuscript). Our results show that TKO CD4⁺ T cells exhibited a significantly increased frequency of IL-17A⁺ cells under pathogenic conditions. Compared with the WT mice, the production of IL-17A slightly decreased in DKO CD4⁺ T cells, suggesting that overproduced IL-4 inhibits IL-17A expression. We have added description about this part in the revised manuscript (page 11, lines 219-223).

Rebuttal Fig. 1 (same as above), The production of cytokines in CD4⁺ T cells cultured in vitro. **a** The procedure for in vitro T cell differentiation. **b** Frequency of IL-4 and IL-17A production in the CD4⁺ T cells from WT, DKO, and TKO mice, which were cultured in vitro as in (a) (key; n = 3 per group). Each symbol represents one individual replicate.

Although the work is generally rigorous, the full set of data represented in the figures is often not clear. Many of the panels say “n=#,” but it's not clear if n refers to biological replicates from a single experiment vs independent experiment. Some figures do not

mention the number of biological replicates used. This should be clarified in the figure legends.

We appreciate this observation. The relevant figure legends have been clarified in the revised manuscript.

=====

Reviewer #2 (Remarks to the Author)

Building upon their previous results (Aki D et al., Nat Immunol 2018), Zhao et al. investigated the roles of Itch and WWP2 in CD4+ T cells within the context of neuroinflammation. Utilizing an extensive set of genetic mouse models and in vitro experiments, the authors found that MOG35-55-specific 2D2-TCR transgenic mice lacking Itch and WWP2 in CD4+ T cells exhibited atypical EAE symptoms, with immune responses characterized by a Th2-driven pathology, primarily mediated by IL-4. Concerning downstream signaling of IL-4, the authors propose that activation of the IL-4 signaling pathway alone is sufficient for inducing the phosphorylation of STAT5. Furthermore, they systematically examined the function of STAT5 in Th17 cell differentiation, concluding that by enhancing the expression of Blimp1 and c-Maf, STAT5 suppresses the initiation of Th17 cell differentiation. As a clinical link to human disease, the authors re-analyzed skin sample datasets from dupilumab-associated psoriasis in individuals with atopic dermatitis and confirmed the upregulation of genes related to pathogenic Th17 cells.

This well-written manuscript presents a thorough analysis of the interaction between Th2 and Th17 differentiation, providing novel insights into the role of Th2-mediated immune responses in the context of autoimmunity. The authors have addressed a clinically relevant issue, and the methodology appears sound. However, a few points require clarification:

We thank this Reviewer for the complimentary remarks in recognizing the quality of our study.

Major Criticism:

- In the introduction, the authors inaccurately describe the current understanding of the pathophysiology of multiple sclerosis (MS) and experimental autoimmune encephalomyelitis (EAE). Specifically, the authors assert that Th17 cells are the "key drivers of MS progression" (page 3, lines 41-22), which does not align with the current consensus on MS-related disease progression (Kuhlmann T et al., Lancet Neurol 2022). Additionally, they claim that "In the central nervous system (CNS), TH17 cells secrete the

inflammatory cytokines IL-17, IFN- γ , and GM-CSF, which recruit polymorphonuclear cells from peripheral tissues to the CNS and lead to neuroinflammation" (page 3, lines 42-44). However, polymorphonuclear cells constitute only a minor population in MS lesions (Lassmann H, Cold Spring Harb Perspect Med 2018). Furthermore, the authors state that "TH2 cells are also implicated in the pathogenesis of MS" (page 3, lines 44-46). To date, convincing evidence for a significant role of Th2 cells has only been demonstrated in the context of EAE, not in human MS (Attfield KE et al., Nat Rev Immunol 2022). We recommend that the authors frame EAE as an appropriate model for studying the differential roles of T helper cells in shaping the autoimmune response, rather than focusing on MS pathophysiology, which is neither correctly described nor revisited in the discussion.

We appreciate the thorough comment of our manuscript. We have carefully considered the raised issues and have made corrections to address the inaccuracies in the descriptions. As the reviewer mentioned, the critical focus of this project was on the function of helper T cells in autoimmune neuroinflammation (EAE) and the trans-differentiation between Th2 and Th17 cells, the description of human MS may not be appropriate. We agree that our original assertion oversimplified the complex pathophysiology of MS. As recommended, we focused on the description about the dysregulation of T cells in EAE rather than MS. We added the description of T cells in EAE in Introduction part (page 3, lines 38-49).

Additional, as suggested, we have added in the Discussion part about the usefulness of EAE and its limitations (page 23, lines 500-505).

- Throughout the methods section, the authors do not consistently specify whether experiments were conducted using 2D2 T cells (e.g., pages 22-24). At times, it is unclear when 2D2 cells were employed. Please clarify when 2D2 cells are used to provide a cohesive statement that acknowledges their use throughout the manuscript, while also noting any exceptions.

We thank this reviewer for the valuable comments. Most of the studies used 2D2 T cells. To further clarify this issue, in the methods section, we have added T cell lines statement in the revised manuscript (page 25, lines 535-537).

- Additionally, it is crucial to ascertain whether the observed immune phenotype is exclusively mediated by 2D2 cells or also involves bystander cells. What is the percentage of 2D2 cells in the analysis, as shown in Figure 1D-F, Figure 2A-D, Figure 2G/H and Figure 3D/E. It is also unclear how many of the transferred cells are 2D2 cells (Figure 2D). Could these be measured using appropriate antibodies against the TCR alpha and beta chains or MOG-tetramers?

We thank the reviewer for highlighting this essential methodological consideration. Consistent with prior findings (Bettelli E et al., J. Exp. Med. 2003), transgenic $V\alpha 3.2^+ V\beta 11^+$ T cells represented >98% of $CD3^+$ T cells in 2D2 transgenic mice. In our analysis, we quantified the proportion of $V\alpha 3.2^+ V\beta 11^+ CD4^+$ 2D2 cell among these four groups and observed that more than 90 percent of $CD4^+$ T cells are 2D2 cells (Rebuttal Fig. 5). These findings indicate that neuroinflammation in DKO and TKO 2D2 mice is specifically mediated by 2D2 cells, not bystander C57BL/6 cells.

Rebuttal Fig. 5, Flow cytometry plots show $V\alpha 3.2$ and $V\beta 11$ co-expression of $CD4^+$ T cells in spleen.

- In Figure 6D, the authors present an immunoblot analysis of phosphorylated and total JAK3. A quantification using densitometry would be good.

We thank this reviewer for the kind notice. An analysis of grayscale values for various proteins in WB has been incorporated into Fig. 6d, and corresponding updates were made to the figure legend and Methods part (page 29, line 624; page 45, lines 958-959).

Minor Criticism:

- To investigate whether the effect of IL-4 on the Th2 to Th17 transition is tissue-specific, the authors analyzed cytokine production in $CD4^+$ T cells from the colon and spleen

(pages 8-10). However, given that other organs, such as the lung and skin, are commonly affected by Th2-mediated diseases (e.g., asthma and atopic dermatitis), it would be valuable to know whether these organs were also analyzed in the study.

We thank the reviewer for this insightful suggestion regarding the tissue-specific effects of IL-4 on Th2-to-Th17 plasticity. We agree that extending our analysis to organs beyond the colon and spleen—particularly those implicated in Th2-driven pathologies, such as the lung (asthma) and skin (atopic dermatitis)—would strengthen the study's translational relevance. To address this point, we conducted additional experiments using lung and skin samples from our mouse models. These data are now included in the Rebuttal Fig. 6 (also supplementary Fig. 3a to i in the revised manuscript) and associated description was added in revised manuscript (page 10, lines 192 to 204). Below, we summarize the key results and their implications.

Lung CD4⁺ T cells:

First, we examined CD4⁺ T cells in the lungs of 6- to 8-week-old WT, *Il4* KO, DKO, and TKO mice. The TKO mice already exhibited classical EAE symptoms. The frequency of CD4⁺ T cells was similar among these four groups (Rebuttal Fig.6a). DKO mice showed enhanced production of IL-4 and GM-CSF in CD4⁺ T cells, while IL-17A production remained unchanged (Rebuttal Fig.6b and c). This suggests that there was no conversion from Th2 to Th17 cells in the lungs of DKO and TKO mice as early as 6-8 weeks of age. Next, we assessed cytokine production in the lungs of mice older than 12 weeks. The production of IL-17A was increased in TKO CD4⁺ T cells compared to DKO mice (Rebuttal Fig.6d and e). We consider this phenomenon in the lung is an indirect effect mediated by the inflammatory environment in older mice. It may be due to activated Th17 cells migrating to the lung, rather than lung-resident T cells recognizing the MOG antigen and differentiating into Th17 cells.

Skin CD4⁺ T cells:

We also analyzed the cytokine levels in the postauricular skin of WT, DKO, and TKO mice. The production of cytokines associated with type 2 inflammation, such as IL-4, IL-13, and IL-25, was increased in DKO mice compared to the WT group (Rebuttal Fig.6f to h). Additionally, IL-17A production was slightly increased in TKO mice compared to DKO

mice and was comparable to the WT group (Rebuttal Fig.6i).

Rebuttal Fig. 6, Cytokine production in lung and skin from WT, *I/4* KO, DKO, and TKO mice. **a** Flow cytometry of lung CD4⁺ T cells from 6- to 8-week-old WT, *I/4* KO, DKO, and TKO mice. **b, c** Flow cytometry plots illustrating the production of IL-4, GM-CSF, IL-17A, and IFN- γ by CD4⁺ T cells in the lungs of mice described in (a). **d, e** Flow cytometry plots showing the production of IL-4, GM-CSF, IL-17A, and IFN- γ by CD4⁺ T cells in the lungs of 12-week-old WT, *I/4* KO, DKO, and TKO mice. **f to i** Relative mRNA expression levels

of *Il4* (f), *Il13* (g), *Il25* (h), *Il17a* (i) in the postauricular skin of mice as in (d, e).

We thank the reviewer for highlighting the importance of tissue context in Th2/Th17 balance. Our new data confirm that the phenomenon of Th2-to-Th17 cell trans-differentiation predominantly occurs within the CNS. This observation may be linked to inflammatory environments, indicating the critical influence of microenvironmental cues on T-cell plasticity. Additionally, this phenomenon could be attributable to the use of 2D2 mice in our experimental model, underscoring the importance of TCR-mediated antigen recognition. We hope that these revisions have fully addressed the raised concerns.

- Considering previous data on the role of IL-4 in the context of neuroinflammation (Ponomarev ED et al., J Neurosci 2007), the authors tend to overstate the novelty of their findings in some instances. We advise them to moderate the wording, e.g., on page 19, lines 407-408 ("Unexpectedly, deletion of IL-4 exacerbated EAE symptoms and led to the development of classical EAE").

We appreciate this kind notice and have thus made changes in the revised manuscript to make the wording more appropriate (page 21, line 438). As the reviewer commented that previous studies have documented the role of IL-4 in neuroinflammation, we initially hypothesized that additional IL-4 deletion would ameliorate EAE symptoms in DKO mice. However, the fact is contrary to our speculation: the absence of IL-4 leads to a shift from Th2 to Th17 cell trans-differentiation, thereby promoting the occurrence of more severe classical EAE symptoms.

=====

Reviewer #3 (Remarks to the Author):

Reviewer #4 (Remarks to the Author):

The manuscript entitled “The E3 ligases Itch and WWP2 regulate autoimmune neuroinflammation by controlling Th2 to Th17 cell conversion via interleukin-4-STAT5 axis” by Zhao et al investigates the roles of Itch, WWP2, and IL-4 in experimental autoimmune encephalomyelitis (EAE) using genetically deficient mouse models. The authors showed that deletion of Itch and WWP2 (DKO) leads to atypical EAE, characterized by neuroinflammation. Further deletion of the Il4 gene in the DKO mice (creating TKO mice) resulted in the conversion of Th2 cells to Th17 cells, driving the development of classical EAE. Mechanistically, they identify the IL-4R/JAK3/STAT5 signaling pathway as a key regulator that promotes Blimp1 expression, which subsequently inhibits Il17a expression. Although the study sheds some light on the phenomenon of atypical neuroinflammation and its molecular mechanism, I have the following concerns regarding the results presented.

We thank the Reviewers for their thoughtful and insightful comments on our manuscript. The suggestions help us to improve the clarity and presentation of our work, as detailed in the point-by-point responses below.

1. The concept of “atypical EAE” remains unclear. While the authors attribute the neuroinflammation observed in DKO mice to IL-4, the co-expression of IL-4 and GM-CSF by DKO CD4+ T cells raises the possibility that GM-CSF, rather than IL-4, is the primary driver of neuroinflammation. GM-CSF is a well-established contributor to EAE pathogenesis, yet the authors have not explored its role in this phenomenon.

We thank the reviewers for highlighting the need to clarify GM-CSF's pathological role in atypical EAE. Atypical EAE, characterized primarily by ataxia, was first described in 1996. While its pathogenesis is linked to inflammatory changes in the brainstem and cerebellum, the underlying mechanisms remain elusive. It has been reported that CNS-infiltrating neutrophils are critical in the pathogenesis of atypical EAE, producing CXCL2 and recruiting themselves to the brainstem (Yudong Liu et al., J Immunol 2024). Notably, IFN- γ and IL-17 are dispensable for the development of atypical EAE (Mark A Kroenke et al., Eur J Immunol 2010; Ti-Ara J Turner et al., J Immunol 2023).

Supporting our findings, John R. Lukens et al. demonstrated that dysregulated IL-4 production contributes to atypical EAE (John R. Lukens et al., *Immunity* 2015). While their work linked neuroinflammation to neutrophil infiltration, our data revealed comparable neutrophil frequencies and numbers in DKO and WT mice (supplementary Fig. 1). Moreover, we observed that eosinophils represented an increased fraction of the infiltrating myeloid cell population in DKO mice. This aligns with clinical observations, eosinophilic vasculitis was also detected in patients with MS (Raphael Schneider et al., *American Academy of Neurology* 2015; Kongsak Tanphaichitr et al., *JAMA Neurology* 1980). Collectively, we hypothesize that Th2 cell-mediated type 2 immunity orchestrates eosinophil recruitment and pathogenic antibody production via IL-4, thereby promoting atypical EAE.

Since GM-CSF can be transiently expressed by several Th subsets (Wanqiang Sheng et al., *Immunome Res* 2015), we propose that elevated GM-CSF production in DKO cells may be related to the increased IL-4-induced STAT5 phosphorylation. GM-CSF is a well-established mediator of neuroinflammation, recruiting circulating myeloid cells and activating phagocytes, especially microglia, in the CNS (Burkhard Becher et al., *Immunity* 2016; Sabine Spath et al., *Immunity* 2017; Eugene D Ponomarev et al., *J Immunol* 2017). Notably, our data demonstrate that both the proportion and number of microglia are significantly increased in the CNS of DKO mice, although the role of microglia in atypical EAE remains controversial (Alejandro Montilla et al., *Cell Death Dis* 2023; Stephen J. Rubino et al., *Nat Commun* 2018). The potential roles of CNS-elevated GM-CSF in DKO mice, particularly regarding neutrophil infiltration and microglia activation, need further exploration. Obviously, this is beyond the scope of the current work. We indeed appreciate the reviewer's comment and further discussed this issue in the Discussion section of the revised manuscript (pages 21-22, lines 454-461).

2. Although the mechanisms underlying Th2-to-Th17 plasticity identified in this study are novel, the advancement of knowledge is limited because the concept of CD4+ T cell plasticity is already well established in the field.

We thank the reviewers to raise the novelty issues. First, we appreciate the reviewer's comment that our findings on Th2 to Th17 plasticity are novel. Second, regarding the general advancement of knowledge in this field, we provide the following reasonings:

As we have already known, since Coffman and Mossman first classified CD4⁺ T cells into Th1 and Th2 subsets based on cytokine profiles in 1986, there has been a rapidly expanding literature surrounding CD4⁺ T cell subsets and making the case for broad plasticity among these subsets. Whereas we take for granted the general theory for Th cell subset plasticity, key spatial-temporal questions remain unresolved: e.g., does the plasticity prefer in a certain direction? or the bifurcation occurs at any stage of T cell differentiation? Importantly, does such conversion bear biological/clinical relevance?

Specifically, Th17 cells have been extensively studied and are considered to have a great degree of plasticity, capable of converting into Th1-like or Treg-like Th17 cells by secreting cytokines such as IFN γ and IL-10. Recent findings have shown their conversion into T follicular helper (Tfh) cells. On the other hand, Th2 cells, as the major players in type 2 immune responses, are widely regarded as a terminally differentiated and stable subset. Currently, apart from studies exploring the association between Th9 and Th2 cells, research on the conversion of Th2 cells to other subsets remains sparse, particularly regarding the relationship to Th17 cells. Prior work revealed that under *in vitro* culture conditions, classical Th2 effector/memory cells had the potential to produce IL-17 after stimulation with proinflammatory cytokines IL-1 β , IL-6, and IL-21 (Yui-Hsi Wang et al., J. Exp. Med. 2010). In 2022, another group demonstrated that ROR γ t deletion drives the conversion of mature Th17 cells into Th2-like cells (Xinxin Chi et al., Sci. Adv. 2022). However, both studies demonstrated terminal-stage T cell plasticity. In our manuscript, we demonstrated that during early-stage of naïve T cell differentiation, loss of IL-4 directs the T cells to differentiate from Th2 into Th17 lineage. Moreover, given the mutual antagonism between Th2 and Th1 cells during differentiation, it would be logical to hypothesize that IL-4 deficiency promotes a shift from Th2 to Th1 cells, thereby enhancing IFN- γ production. Unexpectedly, however, our findings reveal a novel pathway for Th2 cell conversion. More importantly, clinical cases have been reported that patients treated for atopic dermatitis with the IL-4R α inhibitor may experience the onset of psoriasis (cited in the manuscript). However, the underlying mechanisms remained unclear. Our work thus provides molecular insights relevant to human diseases.

Therefore, the current study provides two new findings: (1) The IL-4/JAK3-STAT5 axis acts as a checkpoint limiting Th2-to-Th17 conversion, and (2) IL-4 governs Th2/Th17

plasticity during the initial activation of naïve T cells in inflammatory environment. Such findings advance our knowledge and provide new insights into the “text-book” concept of T helper cell plasticity. Notably, these discoveries elucidate paradoxical clinical outcomes of biologics targeting IL-4/IL-13 and provide actionable strategies to optimize JAK/STAT modulation across immune disorders. To further address the raised issue, we added more discussion on page 23, lines 485-491 in the Discussion part.

In sum, we do appreciate the reviewer’s positive comments about our work, and at the same time, the philosophic point about the general theory of T helper cell plasticity. Obviously, the plasticity issue is far beyond the scope of this manuscript or probably any other single paper. We sincerely believe that the above-described responses have sufficiently addressed the reviewer’s concerns.